

# dS₂ as excitation of AdS₂

**Florian Ecker**[1⋆], **Daniel Grumiller**[1†] **and Robert McNees**[2‡]

**1** Institute for Theoretical Physics, TU Wien,
Wiedner Hauptstrasse 8-10/136 A-1040 Vienna, Austria
**2** Department of Physics, Loyola University Chicago, Chicago, IL, USA

⋆ fecker@hep.itp.tuwien.ac.at , † grumil@hep.itp.tuwien.ac.at , ‡ rmcnees@luc.edu

## Abstract

We introduce a family of 2D dilaton gravity models with state-dependent constant curvature so that dS₂ emerges as an excitation of AdS₂. Curiously, the strong coupling region corresponds to the asymptotic region geometrically. Apart from these key differences, many features resemble the Almheiri–Polchinski model. We discuss perturbative and non-perturbative thermodynamical stability, bubble nucleation through matter shockwaves, and semiclassical backreaction effects. In some of these models, we find that low temperatures are dominated by AdS₂ but high temperatures are dominated by dS₂, concurrent with a recent proposal by Susskind.



# 1  Introduction

Dilaton gravity in two dimensions (2D) provides infinitely many toy models for classical and quantum gravity, black holes, and holography. Some of the attractiveness of 2D dilaton gravities are the universal features that apply to all models, regardless of the choice of the dilaton potential $V\left(X, (\partial X)^2\right)$ in the bulk action

$$I[g_{\mu\nu}, X] = \frac{1}{2\kappa^2} \int_M \mathrm{d}^2 x \sqrt{-g}\left(XR + 2V\left(X, (\partial X)^2\right)\right). \tag{1}$$

For instance, all models (1) can be reformulated as a Poisson sigma model [1, 2], a specific topological gauge theory. Moreover, all classical solutions can be obtained globally for all models [3–5], see also [6, 7] and refs. therein.

However, quite often crucial insights and technical advances rely on specific models. For example, the Jackiw–Teitelboim (JT) model [8, 9] features prominently in AdS$_2$ holography and quantum gravity applications especially in the past decade, see e.g. [10–19]. Similarly, a version of the Callan–Giddings–Harvey–Strominger (CGHS) model [20] arose recently in the contexts of flat space and near horizon holography, see e.g. [21–30].

In the present work, we focus on a specific 2-parameter family of models (1) that exhibits unique features, by choosing

$$V\left(X, (\partial X)^2\right) = a^2 X^3 + a^2 b X^2 + \frac{(\partial X)^2}{X}, \qquad a, b \in \mathbb{R}. \tag{2}$$

As will be shown in our paper, all solutions of these models have constant curvature, but unlike the JT or CGHS models, the magnitude and sign of the curvature are state-dependent. Such a model contains as part of its solution space locally AdS$_2$, locally flat, and locally dS$_2$ spacetimes, so that one can view dS$_2$ as an excitation of AdS$_2$ and transitions between positively and

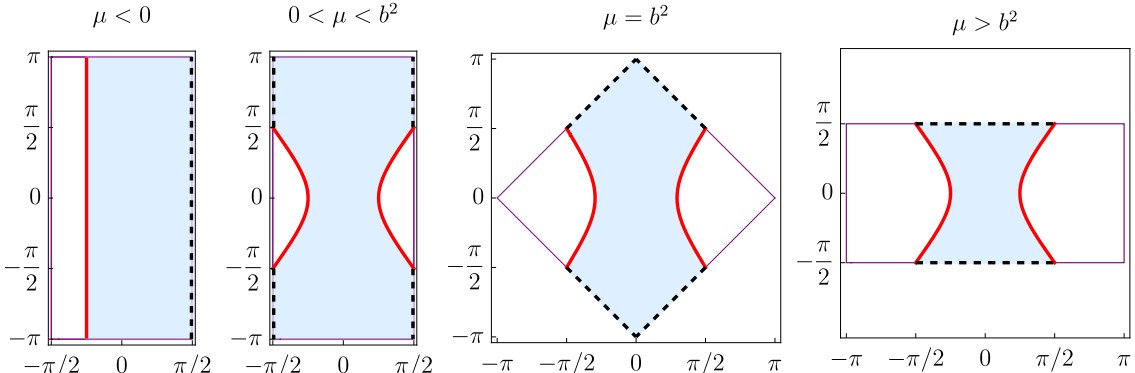

Figure 1: These conformal diagrams are explained in Figure 3. They show a progression from AdS$_2$ (left two plots) via Minkowski$_2$ to dS$_2$ (right plot).

negatively curved spacetimes can arise, in a sense that we are going to make precise in the body of the paper. As a preview, we sketch the conformal diagrams for solutions in a model with negative $b$ and different values of a mass parameter $\mu$ in Figure 1. The main goal of our work is to define and study this model, including classical, thermodynamical, and quantum aspects.

One of the take-away slogans is that most results for this model are opposite to usual expectations. This is so because the weak coupling region ($X \to \infty$) does not turn out to be the asymptotic region geometrically, but rather the center of spacetime; conversely, the strong coupling region ($X \to 0$) is not geometrically a center or singularity, but rather corresponds to the asymptotic region.

We summarize here some additional key results, together with the organization of our paper:

- Classically, we find that dS$_2$ lies at the high-energy end of the spectrum, while AdS$_2$ lies at the low-energy end of the spectrum. Minkowski$_2$ arises in between, for infinite fine-tuning of the energy. See section 2, where we formulate the theory and describe all its classical solutions.

- In the canonical ensemble, for a given temperature $T$, in addition to a unique state with horizon we find a continuum of states without horizon. See section 3, where we define the canonical ensemble associated with our model.

- Relatedly, in the canonical ensemble for models with negative parameter $b$ the dominant state is AdS$_2$ at low temperatures and dS$_2$ at high temperatures. Together with (non-)perturbative stability considerations, this concludes section 3.

- The model remains exactly solvable classically in the presence of scalar matter. We consider specifically matter shockwaves that generate bubble nucleation of spacetime with different curvature and display the associated Penrose diagrams in section 4.

- We quantize the scalar matter fields on a fixed background and take into account back-reactions, leading to numerous subtleties that we address in section 5.

- For a certain (dilaton-dependent) choice of the path integral measure for the scalar field, the theory remains exactly solvable semiclassically. The transition from a stable AdS$_2$ state at low temperature to a stable dS$_2$ state at high temperature remains a feature semiclassically. See the end of section 5.

In addition, a short summary of key results appears at the beginning of each section.

## 2 Vacuum theory

In this section, we describe a 2D dilaton gravity model that yields solutions with state- dependent constant curvature. The curvature is determined by an integration constant and may be positive, zero, or negative for different solutions in the same model. Like many other dilaton gravity models, the formulation of the theory includes a boundary condition that requires the dilaton to take a large value $X_c$ on a regulating surface. This surface is later removed via the limit $X_c \to \infty$. The equations of motion (EOM) for the theory map large values of the dilaton to points deep in the interior of spacetime, allowing solutions with qualitatively different spacetime asymptotics.

### 2.1 Action, boundary conditions, and vacuum equations of motion

Our starting point is dilaton gravity coupled to conformal matter in 2D, with a kinetic term and potential for the dilaton that lead to solutions with state-dependent constant curvature.[1] The model is initially defined on a 2D manifold $M$ with boundary $\Sigma$. The boundary serves as a regulator that allows us to set up a proper variational principle, but it will eventually be removed via a limiting procedure that recovers the full spacetime. The dilaton is denoted by $X$, the metric on $M$ is $g$, and $f$ is a minimally coupled scalar field. In Lorentzian signature the action is

$$\Gamma = \frac{1}{2\kappa^2} \int_M d^2 x \sqrt{-g} \left( X R + \frac{2}{X} (\nabla X)^2 + 2 a^2 X^3 + 2 a^2 b X^2 - \frac{1}{2} (\nabla f)^2 \right) \tag{3}$$
$$+ \frac{1}{\kappa^2} \int_\Sigma dx \sqrt{-h} \left( X K - \mathcal{L}_{\mathrm{ct}}(X, \partial_\parallel X) \right).$$

Here $R$ is the scalar curvature, $h$ is the metric induced on $\Sigma$, and $K$ is the extrinsic curvature of $\Sigma$ embedded in $M$. The parameters $a$ and $b$ are constants; $a$ has units of $(\text{length})^{-1}$ while $b$ is dimensionless. Later on, we shall fix $a$ to a convenient value and consider models corresponding to different values of $b$. The boundary term $\mathcal{L}_{\mathrm{ct}}$, which may in principle depend on the dilaton and its derivative along $\Sigma$, ensures a well-defined variational principle in the limit where the regulating surface $\Sigma$ is removed to infinity [31]. Finally, we include but do not explicitly write out a topological Einstein–Hilbert term in the action so that the effective gravitational coupling remains finite and small over the range $0 \le X < \infty$.

To complete the definition of the theory we must specify boundary conditions for the fields. In higher dimensional gravitational theories this often involves a particular choice of coordinate frame for describing their asymptotic behavior. For example, one might use Schwarzschild-like coordinates where spatial infinity corresponds to $r \to \infty$, and the different components of the metric are required to fall off or grow as particular powers of $r$. An alternative is to use a coordinate-independent definition [32–36], but in either case, a specific notion of spacetime asymptotics is explicit in the field configurations allowed by the theory. Here we require that the dilaton takes a large, fixed value $X_c$ at $\Sigma$ independent of any particular choice of spacetime coordinates. In the Euclidean version of the theory, we also fix the proper period of the Euclidean time at $\Sigma$ to the value $\beta_c$. The boundary conditions can be stated as

$$X \Big|_\Sigma = X_c, \qquad \beta \sqrt{h} \Big|_\Sigma = \beta_c. \tag{4}$$

The limiting procedure mentioned above, which removes the regulating boundary $\Sigma$, is achieved by taking $X_c \to \infty$. In models like the ones considered in [31], the coordinate

---

[1]See Appendix A for a derivation.

dependence of $X$ is such that this procedure can be thought of as removing $\Sigma$ to spatial infinity. But in the present model, the kinetic term for the dilaton leads to large values of the dilaton being mapped to points deep in the interior of spacetime. The boundary conditions at $\Sigma$ and the limit $X_c \to \infty$ ensure that all solutions of the theory have this region in common, but exhibit different spacetime asymptotics. Thus, we encounter (with a few caveats) locally AdS$_2$, Minkowski, and dS$_2$ spacetimes as solutions of a single model.

The boundary condition on the dilaton requires $\Sigma$ to be an isosurface of $X$, and hence derivatives of $X$ along $\Sigma$ vanish. In that case, the boundary counterterm for this model is

$$\mathcal{L}_{\mathrm{ct}}(X) = \sqrt{a^2 X^4 + 2 a^2 b X^3}\,. \tag{5}$$

With this boundary term, the action has a well-defined variational principle for field configurations with the same $X \to \infty$ behavior as solutions of the EOM.[2] A detailed discussion of the variational problem can be found in [31].

Solutions with non-zero matter fields are considered in section 4. For now we set $f = 0$ and focus on solutions involving only the dilaton and metric. The EOM obtained from (3) with $f = 0$ are

$$\nabla_\mu \nabla_\nu X - g_{\mu\nu} \nabla^2 X - \frac{2}{X} \nabla_\mu X \nabla_\nu X + \frac{1}{X} g_{\mu\nu} (\nabla X)^2 + g_{\mu\nu} \left( a^2 X^3 + a^2 b X^2 \right) = 0\,, \tag{6}$$

$$R + \frac{2}{X^2} (\nabla X)^2 - \frac{4}{X} \nabla^2 X + 6 a^2 X^2 + 4 a^2 b X = 0\,. \tag{7}$$

We consider two choices of coordinate gauge when analyzing these equations. In this section and the next we focus on static solutions written in Schwarzschild-like coordinates

$$ds^2 = -\xi(X)\, dt^2 + \frac{1}{\xi(X)}\, dr^2\,, \tag{8}$$

where the dilaton is a function of $r$. It is always possible to express vacuum solutions in this form because the EOM imply a Killing vector with orbits that are isocurves of $X$, which we denote by $\partial_t$. Schwarzschild gauge is useful for establishing a few basic properties of solutions and analyzing the thermodynamics of our model. Beginning in section 4 we frequently work in conformal gauge[3]

$$ds^2 = -e^{2\omega(x^+, x^-)}\, dx^+ dx^-\,, \tag{9}$$

where the dilaton depends of both $x^+$ and $x^-$. This will be useful for incorporating matter, connecting our results to analyses of related models, and studying backreaction. For static solutions the fields depend only on the difference $x^+ - x^-$, and the functions appearing in (8) and (9) are related by

$$\omega = \frac{1}{2} \ln(4\xi)\,. \tag{10}$$

Conformal gauge (9) is preserved by residual diffeomorphisms $x^\pm \to W^\pm(x^\pm)$.

## 2.2 All classical solutions

The most surprising feature of our model is that solutions have state-dependent constant curvature and exhibit different spacetime asymptotics. This is most easily seen in Schwarzschild

---

[2]Since we always consider the $X_c \to \infty$ limit it is sufficient to work with the first three terms in the large-$X$ expansion of (5): $\mathcal{L}_{\mathrm{ct}} \simeq a X^2 + a b X - a b^2/2$.

[3]In later sections we also use the Euclidean analogue of lightcone coordinates $x^\pm$ with $t \to i\tau$.

gauge (8), where the EOM (6) and (7) integrate to

$$\partial_r X = \pm a X^2 \,, \tag{11}$$

$$\xi(X) = 1 + \frac{2\,b}{X} - \frac{2\,M}{X^2} \,. \tag{12}$$

The integration constant $M$ is essentially the energy of our state, a notion we shall make explicit in section 3. Integrating (11) yields $X = \mp(a\,r + c)^{-1}$, with a constant $c$ that can be absorbed by shifting the origin of $r$. Since our solutions must include the region $X \to \infty$ and avoid the strong coupling region at negative $X$, we fix (without loss of generality) $a > 0$ and take the minus sign in (11) to obtain

$$X = \frac{1}{a\,r} \,, \qquad r \geq 0 \,. \tag{13}$$

With this choice $X \to \infty$ corresponds to $r \to 0$, and $r$ takes values in an interval $0 \leq r < r_{\mathrm{max}}$. The norm $\xi$ of the Killing vector $\partial_t$,

$$\xi = 1 + 2\,a\,b\,r - 2\,M\,a^2\,r^2 \,, \tag{14}$$

yields the curvature

$$R = -\partial_r^2 \xi = 4\,a^2\,M \,. \tag{15}$$

As promised, these solutions have constant curvature determined not by the boundary conditions of the theory but by an integration constant $M$, i.e., state-dependent curvature. There is no condition on the sign of $M$, so solutions with positive, zero, or negative curvature are allowed. For the model with $b = 0$ one immediately recognizes $\mathrm{AdS}_2$ ($M < 0$), Minkowski$_2$ ($M = 0$), and the static patch of $\mathrm{dS}_2$ ($M > 0$) in (14). For models with non-zero $b$, there are additional possibilities.

The condition $\xi \geq 0$ determines the upper end of the interval $0 \leq r < r_{\mathrm{max}}$ according to whether or not $\xi$ has a zero (horizon) at some finite $r_h > 0$. For simplicity, we set $a^2 = 1$ and consider the allowed range of $r$ for solutions of models labeled by the parameter $b$ in the potential. In general, any model admits solutions both with and without horizon. In the next section, we shall turn to the Euclidean theory and consider which of those states are compatible with the finite temperature boundary condition (4) that fixes the proper period of the Euclidean time at $X \to \infty$ ($r \to 0$). For now, we describe solutions depending on the sign of $b$. It is convenient to first shift the integration constant $M$ by a model-dependent term

$$M = \frac{\mu - b^2}{2} \,. \tag{16}$$

Then the Killing norm $\xi$ and the scalar curvature are given by

$$\xi = (1 + b\,r)^2 - \mu\,r^2 \,, \qquad R = 2(\mu - b^2) \,. \tag{17}$$

For models with $b \geq 0$, solutions with a cosmological horizon exist for $\mu > b^2$. This implies $R > 0$. So all such solutions are locally $\mathrm{dS}_2$. The horizon is at $r_h = (\sqrt{\mu} - b)^{-1}$, and regularity fixes the period of the Euclidean time to the $b-$independent value

$$\beta = -\frac{4\pi}{\partial_r \xi}\Big|_{r_h} = \frac{2\pi}{\sqrt{\mu}} \,. \tag{18}$$

When $\mu = b^2$ the curvature vanishes, and $r$ takes values on the semi-infinite interval $0 \leq r < \infty$ corresponding to $\infty > X > 0$. For $\mu < b^2$ the curvature is negative, and $r$ once again takes

Table 1: Properties of solutions in Schwarzschild gauge. The second column lists the range/value of the mass parameter $\mu$, while the third column shows the sign/value/range of the curvature. The fourth column gives the value of the dilaton at $r_{\text{max}}$, as described in the text. This is positive for solutions with horizon and 0 for solutions with $0 \leq r < \infty$. The fifth column describes the admissible range/specific value of $\beta$.

| $b$ | $\mu$ | $R$ | $X(r_{\text{max}})$ | Allowed $\beta$ |
|---|---|---|---|---|
| | $> b^2$ | $+$ | $\sqrt{\mu} - b$ | $0 < \beta < \frac{2\pi}{b}$ |
| $+$ | $b^2$ | $0$ | $0$ | Any |
| | $< b^2$ | $-$ | $0$ | Any |
| | $> 0$ | $+$ | $\sqrt{\mu}$ | $0 < \beta < \infty$ |
| $0$ | $0$ | $0$ | $0$ | Any |
| | $< 0$ | $-$ | $0$ | Any |
| | $> b^2$ | $+$ | $\sqrt{\mu} + |b|$ | $0 < \beta < \frac{2\pi}{|b|}$ |
| | $b^2$ | $0$ | $2|b|$ | $\frac{2\pi}{|b|}$ |
| $-$ | $0 < \mu < b^2$ | $-2b^2 < R < 0$ | $\sqrt{\mu} + |b|$ | $\frac{2\pi}{|b|} < \beta < \infty$ |
| | $0$ | $-2b^2$ | $|b|$ | $\to \infty$ |
| | $< 0$ | $< -2b^2$ | $0$ | Any |

values in $0 \leq r < \infty$. In the last two cases, $R \leq 0$, the Euclidean solutions are regular for any period $\tau \sim \tau + \beta$.

Models with $b < 0$ have solutions with horizon for $\mu \geq 0$ and hence the curvature $R = 2(\mu - b^2)$ may be positive, zero, or negative. In all cases the horizon is at $r_h = (\sqrt{\mu} + |b|)^{-1}$ and regularity of the Euclidean solution once again requires (18). The solution is non-extremal for $\mu > 0$, while the case $\mu = 0$ ($R = -2b^2$) is the extremal limit $\beta \to \infty$ of a horizon patch in a locally $\text{AdS}_2$ spacetime. Negative values of $\mu$ are solutions with curvature $R < -2b^2$. They have no horizon and $r$ takes values in $0 \leq r < \infty$.

The various solutions and some of their properties for models with different values of $b$ are summarized in Table 1.

Notice that for these solutions the dilaton takes values $X(r_{\text{max}}) < X < \infty$ with $X(r_{\text{max}}) > 0$, corresponding to the Schwarzschild coordinate $r$ in the range $0 \leq r \leq r_{\text{max}}$. This is what one would expect for the dimensional reduction of a higher-dimensional theory, but for an inherently 2D theory a coordinate like $r$ would normally cover a range $-r_{\text{max}} \leq r \leq r_{\text{max}}$. The form of the dilaton (13) prevents us from continuing these solutions to negative values of $r$, since $X \to -\infty$ for $r \to 0^-$. For models with $b = 0$ two copies of the solution can be glued together with the replacement $r \to -r$ for $r < 0$, which is still a solution with constant curvature $R = 4a^2 M$. But for $b \neq 0$ this construction produces a Dirac delta in the curvature at $r = 0$, $R = 4a^2 M - 4ab\,\delta(r)$. Thus, these solutions cover only part of the full conformal diagram associated with $\text{dS}_2$, Minkowski$_2$, or $\text{AdS}_2$. This will be examined in more detail when we construct conformal diagrams in section 4.3. For now we avoid this issue and restrict our attention to the interval $0 \leq r \leq r_{\text{max}}$.

## 2.3 Weyl-rescalings

There are a few obvious connections between the model (3) and some well-studied dilaton gravity theories, most notably the Almheiri–Polchinski (AP) model [10].

Implementing a Weyl-rescaling of the metric with the change of variable $g_{\mu\nu} = X^\alpha \hat{g}_{\mu\nu}$, the bulk term in (3) becomes

$$L_M \sim \sqrt{\hat{g}} \left( X \hat{R} - \alpha \hat{\nabla}^2 X + \frac{2+\alpha}{X} \left( \hat{\nabla} X \right)^2 + 2 a^2 X^{3+\alpha} + 2 a^2 b X^{2+\alpha} \right). \qquad (19)$$

Setting $\alpha = -2$ eliminates the dilaton kinetic term. Then, after discarding the total derivative $\hat{\nabla}^2 X$, we recover the AP model [10] with the parameters in their dilaton potential given by $A = 2 a^2$ and $C = -2 a^2 b$. With $b = 0$ this is the usual JT model [8,9], and if we flip the sign of our explicitly positive coefficient $a^2 \to -a^2$ we obtain the "nearly de Sitter gravity" studied by Maldacena, Turiaci, and Yang in [37]. The rescaling with $\alpha = -1$ reproduces the CGHS model [20] when $b$ is set to zero.

Since $g_{\mu\nu} = X^\alpha \hat{g}_{\mu\nu}$ is just a change of variable, this sort of rescaling is often used to justify working in a convenient choice of the conformal frame where the dilaton kinetic term vanishes. One might conclude that the theory we study is simply the AP model rewritten in different variables. It is true that the EOM (6)-(7), rewritten in terms of $\hat{g}_{\mu\nu}$ rather than $g_{\mu\nu}$, are equivalent to the EOM for the AP model. However, there are three important differences between the two models. First, the AP model avoids solutions of the EOM which include the region $X = 0$, since this is taken to be a curvature singularity in a higher dimensional theory. We take an inherently 2D point of view with our model, where the only concern with $X \to 0$ is whether this corresponds to infinite gravitational coupling. Including a topological Einstein–Hilbert term in the action pushes the strong coupling region to some sufficiently large and negative value of $X$ so that solutions including the region where $X = 0$ are permitted alongside other solutions. Second, the actions for the two models differ by non-zero terms arising from integration-by-parts. This does not affect the EOM, but it does affect the value of the on-shell action and thus the free energy. And third, we consider all signs of the parameter $b$, rather than restricting to $b < 0$ (which corresponds to $C > 0$ in [10]). Thus, while there is a simple map between solutions of the AP model and some of the solutions we consider, the two models are not quite the same even classically.

Nevertheless, the map between our EOM (6)-(7) and those of the AP model is useful when constructing solutions with matter, and some of the phenomena considered there have interesting interpretations in our model. So it will sometimes be convenient for us to make the change of variable

$$g_{\mu\nu} = X^{-2} \hat{g}_{\mu\nu}. \qquad (20)$$

This must be done with care in the semiclassical theory (section 5) where Weyl invariance of the classical theory is violated.

## 3   Thermodynamics

For the Euclidean version of our theory we define a canonical ensemble as in [31], with boundary conditions at $X_c \to \infty$ that fix the temperature $T = \beta^{-1}$. The ensemble includes a single regular solution with horizon, which is the state of lowest free energy, and a continuum of states without horizon. For models with $b \geq 0$ the solution with horizon is locally $dS_2$, while for models with $b < 0$ it is locally $AdS_2$ at temperatures $2\pi T < |b|$ and locally $dS_2$ at temperatures $2\pi T > |b|$. In $b > 0$ models this picture may break down at low temperatures due to non-perturbative effects from the full continuum of horizonless states. But the spectrum of models with $b < 0$ exhibits a finite gap in the free energy that probably protects them from such effects.

### 3.1 Canonical ensemble

The canonical ensemble for general dilaton gravity models was studied in [31]. Here we summarize the relevant arguments as they apply to the Euclidean version of the model described in the previous section.

Construction of the canonical ensemble begins by interpreting the regulating surface $\Sigma$ as a cavity wall that couples the system to a thermal reservoir. This reservoir fixes both the local proper temperature $T_c = \beta_c^{-1}$ and a conserved dilaton charge that we take to be $X_c$. Classical solutions of the theory are saddle points of the Euclidean path integral, with the on-shell action for a solution related to its Helmholtz free energy by $\Gamma_c = \beta_c F_c$ (the subscript indicates dependence on the quantities $X_c$ and $\beta_c$ fixed at $\Sigma$). We are interested in models with boundary conditions such that the classical saddle points include a solution with horizon at $X_h > 0$. For the solution with horizon to be stable in the canonical ensemble, the Gaussian integral over fluctuations around this state in the path integral should converge. In a general dilaton gravity theory, one can always find values of $X_c$ in a neighborhood of $X_h$ such that this is true, but this neighborhood may not include the limit $X_c \to \infty$ that decouples the system from the reservoir. However, in the case of our model, the Gaussian integral over fluctuations converges even in this limit. Let us now consider the various saddle points of the Euclidean action and their properties as the system is decoupled from the reservoir.

For the Euclidean theory, any finite values of the boundary conditions $\beta_c > 0$ and $X_c \gg 1$ at $\Sigma$ allow two types of saddle points. Both can be written in the form

$$ds^2 = \xi(X)\, d\tau^2 + \frac{1}{\xi(X)}\, dr^2, \qquad \tau \sim \tau + \beta, \tag{21}$$

with the Killing norm

$$\xi(X) = \left(1 + \frac{b}{X}\right)^2 - \frac{\mu}{X^2}, \tag{22}$$

and the dilaton given by

$$X = \frac{1}{r}. \tag{23}$$

Then $\xi_c = \xi(X_c)$ is the induced metric on $\Sigma$, and the boundary condition (4) is $\beta_c = \beta \sqrt{\xi_c}$. Note that $\lim_{X_c \to \infty} \beta_c = \beta$.

The first type of a saddle point occurs for an isolated value of $\mu$ and corresponds to the Lorentzian signature solution with horizon at $X_h = \sqrt{\mu} - b > 0$. The regularity condition (18) fixes $\beta$ (and hence $\mu$) in terms of $X_c$ and $\beta_c$

$$\beta = \frac{\sqrt{\beta_c^2 X_c^2 + 4\pi^2}}{X_c + b}. \tag{24}$$

The condition $X_h > 0$ implies $\beta < 2\pi/b$ for models with $b > 0$, but any $\beta > 0$ is allowed for models with $b \leq 0$.

The second type of saddle point is a state of the form (22) with $\xi(X) > 0$ on $0 \leq X \leq X_c$. Positivity of $\xi$ on the full range of $X$ implies $-\infty < \mu < b^2$ in models with $b \geq 0$, and $-\infty < \mu < 0$ for models with $b < 0$. Since $\xi(X) \neq 0$ there is no regularity condition on the period $\beta$, and the boundary condition is satisfied for

$$\beta = \frac{\beta_c X_c}{\sqrt{(X_c + b)^2 - \mu}} > 0. \tag{25}$$

There is a continuum of such states labeled by the value of $\mu$.

Without including the full details of the calculation, the solution with horizon is the state of lowest free energy and dominates the saddle point approximation of the Euclidean path

integral. Following the arguments in [31], stability of this state in the canonical ensemble as the system is decoupled from the reservoir ($X_c \to \infty$) amounts to the condition

$$\lim_{X_c \to \infty} \frac{4\pi^2 \beta_c^2 X_c^2 (b + X_c)}{(\beta_c^2 X_c^2 + 4\pi^2)^{\frac{3}{2}}} > 0, \tag{26}$$

which is always satisfied.[4] In this limit the period $\beta$ for both types of saddle points approaches the value $\beta_c$, which we henceforth refer to as $\beta$. The temperature is related to the period by $T = \beta^{-1}$, so this state is stable against thermal fluctuations in an ensemble defined at any temperature for models with $b \le 0$, and at temperatures $T > \frac{b}{2\pi}$ for models with $b > 0$.

The remaining saddle points are the continuum of horizonless states. In the limit $X_c \to \infty$ the condition (25) describes regular solutions with $\tau \sim \tau + \beta$ for any value of $\mu$ such that $\xi > 0$ on $0 < X < \infty$. These solutions are all at the same temperature $T = \beta^{-1}$ fixed by the boundary conditions. The parameter $\mu$ in (22) may be negative for these states, so to avoid confusion over signs in later discussions we rewrite $\xi$ for the horizonless solutions as

$$\xi = \left(1 + \frac{b}{X}\right)^2 + \frac{(\lambda - b^2)}{X^2}. \tag{27}$$

Then $\lambda$, defined as

$$\lambda = b^2 - \mu, \tag{28}$$

is a continuous parameter that takes positive values $b^2 < \lambda < \infty$ for models with $b < 0$, and non-negative values $0 \le \lambda < \infty$ for models with $b \ge 0$.

## 3.2 Euclidean action and free energy

The holographically renormalized Euclidean action for our model is

$$\Gamma_E = -\frac{1}{2\kappa^2} \int_M d^2x \sqrt{g} \left( X R + \frac{2}{X} (\nabla X)^2 + 2X^3 + 2bX^2 - \frac{1}{2}(\nabla f)^2 \right) \tag{29}$$
$$- \frac{1}{\kappa^2} \int_\Sigma dx \sqrt{h} \left( X K - \sqrt{X^4 + 2bX^3} \right).$$

As before we set the matter fields to zero and work in Schwarzschild gauge (21)-(23).

Let us first evaluate the Euclidean action for the regular solution with horizon, (21)-(23). In that case the integral over $r$ covers the range $r_c \le r < r_h$. The horizon is at $r_h = (\sqrt{\mu} - b)^{-1}$, and the $X_c \to \infty$ limit corresponds to $r_c \to 0$. Individual terms in (29) contain contributions that diverge as $r_c \to 0$, but all such terms cancel.[5] In the $r_c \to 0$ limit, the on-shell action for the solution with horizon is

$$\Gamma_E = -\frac{1}{2\kappa^2} \beta \left( \frac{2\pi}{\beta} - b \right)^2. \tag{30}$$

The on-shell action is negative for any value of $b$ and any finite $\beta$. The Helmholtz free energy $F = \beta^{-1} \Gamma_E$ for the solution with horizon is

$$F = -\frac{1}{2\kappa^2} \left( 2\pi T - b \right)^2, \tag{31}$$

---

[4]This is just the requirement that the specific heat at constant $X_c$ remain positive as $X_c \to \infty$.

[5]In evaluating (29), the outward-pointing unit normal vector at $\Sigma$ used to evaluate $K$ is $n^\mu = -\sqrt{\xi}\, \delta_r^\mu$.

which gives an entropy

$$S = -\frac{\partial F}{\partial T} = \frac{2\pi}{\kappa^2}(2\pi T - b) = \frac{2\pi}{\kappa^2} X_h = \frac{X_h}{4G}. \tag{32}$$

As expected, the entropy reduces to the standard result, namely the dilaton at the horizon divided by $4G$ [38–40]. The internal energy is then equal to the mass parameter,

$$E = F + TS = (\mu - b^2)/2\kappa^2 = M. \tag{33}$$

This result also agrees with the internal energy obtained from the Brown–York quasilocal stress tensor [41] associated with the action (29).

The on-shell action for the continuum of horizonless states takes a qualitatively different form than (30). In that case $r$ covers the range $r_c \leq r < \infty$. Evaluating the action for the solution (27) gives

$$\Gamma_E = \frac{3}{2}\beta\lambda. \tag{34}$$

The free energy for these states is

$$F(\lambda) = \frac{3}{2}\lambda, \tag{35}$$

which is non-negative for models with $b \geq 0$ and positive for models with $b < 0$. Since there is no horizon the free energy does not depend on $T$ and there is no entropy associated with these states.

Comparing (35) with (31), it is clear that the solution with horizon is always the state of lowest free energy. The nature of this solution depends on the sign of $b$ and, for negative $b$, also on the temperature set by the boundary conditions. For models with $b \geq 0$, the state with horizon always has $\mu > b^2$ and corresponds to a locally $dS_2$ solution with $R = 2(\mu - b^2) > 0$. But for models with $b < 0$ the sign of the curvature may be positive, zero, or negative depending on the temperature. In that case there is a solution with horizon for any value of $\beta = T^{-1}$, and the curvature can be written as $R = 2(4\pi^2 T^2 - b^2)$. Thus, for models with $b < 0$ there is a low temperature regime $2\pi T < |b|$ where the state of lowest free energy is a horizon wedge of $AdS_2$, and a high-temperature regime $2\pi T > |b|$ where the state of lowest free energy is the static patch of $dS_2$. At the boundary $2\pi T = |b|$ between the high- and low-temperature regimes the horizon state is locally Minkowski.

### 3.3 Non-perturbative thermodynamical stability

In the canonical ensemble the solution with horizon has negative free energy, while the horizonless states have positive free energy. One would therefore expect the former to dominate the latter in the saddle point approximation. A horizonless state's contribution to the path integral is suppressed relative to that of the horizon state by a factor

$$\mathcal{R} \sim \exp[-\beta\,\Delta F], \tag{36}$$

where $\Delta F$ is the difference in free energy between the two states

$$\Delta F = \frac{3}{2}\lambda + \frac{1}{2}\left(\frac{2\pi}{\beta} - b\right)^2. \tag{37}$$

The difference in free energy is strictly positive, so the relative contribution of the horizonless state is suppressed.

Even though contributions from horizonless states are suppressed, there are infinitely many of them. The saddle point approximation of the path integral should include contributions from all classical solutions, and it is not immediately obvious whether the sum over the full continuum of solutions $\lambda_{\min} \leq \lambda < \infty$ can outweigh the contribution from the state of lowest free energy. This would represent a non-perturbative instability where the interpretation of the Euclidean path integral as describing a canonical ensemble dominated by a stable state with horizon breaks down.

Consider the difference in free energy between the solution with horizon and the lowest-lying horizonless state at $\lambda_{\min}$. For models with $b < 0$ the lowest horizonless state has $\lambda_{\min} = b^2$. Then an instability of this sort is unlikely thanks to a positive gap in $\Delta F$ that is non-zero even in the zero-temperature limit

$$\Delta F = 2\,|b|^2 + 2\pi^2\,T^2 + 2\pi\,|b|\,T > 2\,|b|^2\,. \tag{38}$$

But for models with $b \geq 0$ this gap is not present. The lowest horizonless state in that case has $\lambda_{\min} = 0$, and the allowed boundary conditions are $2\pi\,T > b$. Near the minimum value we can parameterize the temperature as

$$T = \frac{b}{2\pi}\,(1 + \epsilon)\,, \tag{39}$$

and the difference in free energy becomes

$$\Delta F = \frac{1}{2}\,b^2\,\epsilon^2\,. \tag{40}$$

This can be brought arbitrarily close to zero, so for these models, we might expect contributions to the path integral from the full continuum of horizonless states to dominate the contribution from the dS$_2$ state near the minimum temperature.

To make this more precise, consider the cumulative contributions to the Euclidean path integral obtained by integrating $\mathcal{R}$ over the full continuum of horizonless states. Ignoring a possible prefactor in (36) we have

$$\int_{\lambda_{\min}}^{\infty} \mathrm{d}\lambda\,\exp\left[-\beta\,\Delta F\right] = \exp\left[-\frac{1}{2}\,\beta\left(\frac{2\pi}{\beta} - b\right)^2 - \frac{3}{2}\,\beta\,\lambda_{\min} + \ln\left(\frac{2}{3\beta}\right) + \ln c\right]\,. \tag{41}$$

Here $c$ is some constant that reflects an ambiguity in the choice of measure. The contributions from the horizonless states are subdominant to the contribution from the horizon state for boundary conditions $\beta$ that satisfy a positivity condition of the form

$$\mathrm{St}(\beta, b, c) = \frac{1}{2}\,\beta\left(\frac{2\pi}{\beta} - b\right)^2 + \frac{3}{2}\,\beta\,\lambda_{\min} + \ln\left(\frac{3}{2}\,\beta\right) - \ln c > 0\,. \tag{42}$$

The presence of a prefactor in (36) would alter the argument of the first ln term in this function, but we do not expect this to qualitatively change the following conclusions. Let us now consider whether such a stability condition might plausibly be violated.

In models with $b < 0$, boundary conditions with any value $0 < \beta < \infty$ are allowed. The horizonless states have $\lambda > b^2$ so the left-hand side of the stability condition (42) is

$$\mathrm{St}(\beta, -|b|, c) = \frac{2\pi^2}{\beta} + 2\pi\,|b| + 2\,b^2\,\beta + \ln\left(\frac{3}{2}\,\beta\right) - \ln c\,. \tag{43}$$

This is a convex function of $\beta$ on $(0, \infty)$ with a local minimum at $\beta_*$ given by

$$\beta_* = \frac{-1 + \sqrt{1 + 16\,\pi^2\,b^2}}{4\,b^2}\,. \tag{44}$$

Its value at the minimum is

$$\text{St}(\beta_*, -|b|, c) = 2\pi\,|b| + \sqrt{1 + 16\,\pi^2\,b^2} + \ln\left(\frac{3}{8\,b^2}\left(\sqrt{1 + 16\,\pi^2\,b^2} - 1\right)\right) - \ln c\,, \tag{45}$$

which is a monotonically increasing function of $|b| > 0$. This is positive if $c$ is less than a rapidly (exponentially) increasing function of $|b|$, and for all $|b|$ if $c < 3\pi^2\,e$. In that case a stability condition like (42) is satisfied for all boundary conditions $\beta$. If $c$ is large enough such that (45) is negative for some values of $|b|$, the function (43) still becomes positive as $\beta \to 0$ or $\beta \to \infty$. Thus, even for very large values of $c$, all models with $b < 0$ would have high- and low-temperature regimes where the saddle point approximation of the Euclidean path integral describes a stable horizon state in the canonical ensemble, despite contributions from the continuum of horizonless states.

The results are qualitatively similar in models with $b = 0$, where

$$\text{St}(\beta, 0, c) = \frac{2\pi^2}{\beta} + \ln\left(\frac{3}{2}\beta\right) - \ln c\,. \tag{46}$$

As in the $b < 0$ case, this is a convex function of $\beta$ with local minimum at $\beta_* = 2\pi^2$. It is positive for all values of $\beta$ if $c < 3\pi^2\,e$. If $c$ exceeds this value then a stability condition like (42) is not satisfied for some range $\beta_{\text{high}} < \beta < \beta_{\text{low}}$. However, like the case $b < 0$, the function (46) becomes positive as $\beta \to 0$ or $\beta \to \infty$, so the solution with horizon would still dominate over contributions from the continuum of horizonless states in high-temperature ($T > \beta_{\text{high}}^{-1}$) and low-temperature ($T < \beta_{\text{low}}^{-1}$) regimes.

There is a wider range of possibilities for models with $b > 0$. In that case the allowed boundary conditions are $0 < \beta < 2\pi/b$. The continuum of horizonless states begins at $\lambda_{\text{min}} = 0$, so the function appearing in the stability condition is

$$\text{St}(\beta, b > 0, c) = \frac{2\pi^2}{\beta} - 2\pi\,b + \frac{1}{2}\,b^2\,\beta + \ln\left(\frac{3}{2}\beta\right) - \ln c\,. \tag{47}$$

This is positive at $\beta \to 0$, while at $\beta = 2\pi/b$ it takes the value

$$\text{St}\left(\frac{2\pi}{b}, b, c\right) = \ln\frac{3\pi}{b\,c}\,, \tag{48}$$

which is positive for $b\,c < 3\pi$ and negative if $b\,c > 3\pi$. There is a local minimum at $\beta_*$ given by

$$\beta_* = \frac{-1 + \sqrt{1 + 4\,\pi^2\,b^2}}{b^2}\,, \tag{49}$$

where (47) takes the value

$$\text{St}(\beta_*, b > 0, c) = -2\pi\,b + \sqrt{1 + 4\,\pi^2\,b^2} + \ln\frac{\sqrt{1 + 4\,\pi^2\,b^2} - 1}{2\pi\,b} + \ln\frac{3\pi}{b\,c}\,. \tag{50}$$

This may be positive or negative, but it is strictly less than (48). The values at $\beta_*$ and $\beta = 2\pi/b$ are both monotonically decreasing functions of $b$, so there are three possible behaviors for a stability condition like (42) with a given value of $c$. For sufficiently small $b$ both (50) and (48)

are positive, and the stability condition is satisfied for the full range of boundary conditions $0 < \beta < 2\pi/b$. For larger values of $b$, (50) becomes negative but (48) remains positive. Then the stability condition is violated in some interval $\beta_{\text{high}} < \beta < \beta_{\text{low}}$. And for models with $b > 3\pi/c$ both (50) and (48) are negative. In those models the stability condition is violated for all $\beta$ above some critical value: $\beta_{\text{crit}} < \beta < 2\pi/b$. Then the description of the system in terms of a stable horizon state in the canonical ensemble would break down below a critical temperature $T_{\text{crit}} = \beta_{\text{crit}}^{-1}$. For any value of $b > 0$, the model would still satisfy the stability condition at sufficiently high temperatures ($\beta \to 0$).

The general picture, if we assume that the stability condition takes the form (42) with $c \sim \mathcal{O}(1)$, is that the saddle point approximation of the Euclidean path integral for models with $b \leq 0$ is dominated by the solution with horizon for all values of $\beta$, even though there are competing contributions from an infinite number of horizonless states. The same is true for models with $b > 0$ when $b \sim \mathcal{O}(1)$, but at sufficiently large values of $b$ the stability condition is violated below a critical temperature $T_{\text{crit}}$. This critical temperature is larger than the minimum temperature $b/(2\pi)$ that admits a solution with horizon, so there is a range of temperatures

$$b/(2\pi) < T < T_{\text{crit}}, \tag{51}$$

where contributions from the full continuum of horizonless states dominate. In that case, we expect that it is no longer appropriate to interpret the Euclidean path integral as describing a canonical ensemble dominated by a stable state with horizon. To avoid dealing with this non-perturbative instability, we focus on negative $b$ models in the semiclassical discussion in section 5. But before arriving there, we introduce matter in the next section.

## 4 Adding scalar matter

In the previous sections, we considered solutions with the matter field set to zero. Here, we work out general solutions with non-zero matter. The analysis in conformal gauge leads to an interesting interpretation of this model in terms of the dilaton's behavior on a fixed auxiliary AdS$_2$ spacetime. We revisit the properties of vacuum solutions in this context, then construct an explicit solution with matter describing nucleation of a region of different spacetime curvature. Throughout this section we set $\kappa^2 = 1$ for simplicity.

### 4.1 Equations of motion

Including contributions from the matter field, the bulk term in the variation of the action (3) is

$$\frac{1}{2} \int_M d^2x \sqrt{g} \Big[ (\mathcal{E}^{\mu\nu} + T^{\mu\nu}) \delta g_{\mu\nu} + \mathcal{E}_X \delta X + \mathcal{E}_f \delta f \Big]. \tag{52}$$

The EOM for the matter field $f$ is the Klein–Gordon equation

$$\mathcal{E}_f = \nabla^2 f = 0, \tag{53}$$

while $\mathcal{E}_{\mu\nu}$ and $\mathcal{E}_X$ are given by (6) and (7), respectively. There is no matter contribution to the dilaton EOM, but the minimal coupling to gravity gives the stress-energy tensor

$$T_{\mu\nu} = \frac{1}{2} \Big( \partial_\mu f \, \partial_\nu f - \frac{1}{2} g_{\mu\nu} (\partial f)^2 \Big). \tag{54}$$

The Schwarzschild-type coordinates introduced in section 2.2 are useful for quickly establishing the properties of solutions with the matter field set to zero. But to solve the EOM with

non-zero matter we switch to conformal gauge (9). In conformal gauge, the $+-$ component of the stress-energy tensor vanishes, and the resulting EOM are

$$0 = -\partial_+\partial_- X - \frac{1}{2}e^{2\omega}\left(a^2 X^3 + a^2 b X^2\right),\tag{55}$$

$$0 = 8e^{-2\omega}\left(\partial_+\partial_-\omega + \frac{2}{X}\partial_+\partial_- X - \frac{1}{X^2}\partial_+ X\,\partial_- X\right) + 6a^2 X^2 + 4a^2 b X,\tag{56}$$

$$0 = -2\,\partial_+\partial_- f\,.\tag{57}$$

In addition, since the $\pm\pm$ components of the metric (9) are zero, the corresponding components of $\mathcal{E}_{\mu\nu} + T_{\mu\nu} = 0$ enforces the constraints

$$0 = X^2 e^{2\omega}\partial_\pm\left(X^{-2}e^{-2\omega}\partial_\pm X\right) + \frac{1}{2}(\partial_\pm f)^2\,.\tag{58}$$

The matter field $f$ is absent from the dilaton and metric EOM (55), (56), but appears in the constraints (58) and the Klein–Gordon equation (53).

## 4.2  Solutions with scalar matter

The EOM and constraints (55)-(58) can be solved exactly. The matter equation (57) is trivial, with solutions

$$f(x^+, x^-) = f_+(x^+) + f_-(x^-),\tag{59}$$

for arbitrary functions $f_+$ and $f_-$. For the remaining equations, it is useful to first make the change of variable mentioned in section 2.3,

$$\hat{\omega} = \omega + \ln X\,.\tag{60}$$

Then the EOM (55)-(56) are equivalent to the system

$$0 = -\partial_+\partial_- X - \frac{1}{2}a^2 e^{2\hat{\omega}}\left(X + b\right),\tag{61}$$

$$0 = \partial_+\partial_-\hat{\omega} + \frac{1}{4}a^2 e^{2\hat{\omega}}\,,\tag{62}$$

while the constraints (58) become

$$0 = e^{2\hat{\omega}}\partial_\pm\left(e^{-2\hat{\omega}}\partial_\pm X\right) + \frac{1}{2}\left(\partial_\pm f\right)^2\,.\tag{63}$$

These equations have the same form as the equations for the model studied in [10, 13] and can be solved using the same techniques.

After making the change of variable we find an equation for $\hat{\omega}$ that decouples from both the dilaton and matter field. Equation (62) is just the statement that $\hat{g}_{\mu\nu}$ is a metric of constant negative curvature $\hat{R} = -2a^2$. A solution can be written in terms of a pair of monotonic functions $W^+(x^+)$ and $W^-(x^-)$ as

$$e^{2\hat{\omega}} = \frac{4}{a^2}\frac{\partial_+ W^+(x^+)\partial_- W^-(x^-)}{(W^+(x^+) - W^-(x^-))^2}\,.\tag{64}$$

For now, we set $a = 1$ and work in Poincaré coordinates $W^\pm = x^\pm$. Then the conformal factor in the line element is

$$e^{2\hat{\omega}} = \frac{4}{(x^+ - x^-)^2}\,.\tag{65}$$

Writing the dilaton $X$ with an explicit factor of $(x^+ - x^-)^{-1}$ as

$$X = \frac{N(x^+, x^-)}{x^+ - x^-}, \tag{66}$$

the EOM (61) and constraints (63) take the simple form

$$0 = (x^+ - x^-)\, \partial_+ \partial_- N + \partial_+ N - \partial_- N + 2\, b\,, \tag{67}$$

$$0 = \partial_\pm^2 N + (x^+ - x^-)\, T_{\pm\pm}(x^\pm)\,. \tag{68}$$

In the last line, the $\pm\pm$ components of the matter stress-energy tensor are

$$T_{\pm\pm} = \frac{1}{2}\left(\partial_\pm f\right)^2. \tag{69}$$

Solutions of the matter EOM have the form (59), so the $T_{++}$ and $T_{--}$ components are functions of $x^+$ and $x^-$, respectively.

The general solution of (67) and (68) can be written as a vacuum solution with $T_{\pm\pm} = 0$, plus terms that are sourced by the components of the stress-energy tensor:

$$N(x^+, x^-) = N_0(x^+, x^-) + I^+(x^+, x^-) - I^-(x^+, x^-)\,. \tag{70}$$

The constraints (68) require that the vacuum solution $N_0$ satisfy $\partial_+^2 N_0 = \partial_-^2 N_0 = 0$, which can be integrated to give

$$N_0(x^+, x^-) = d_0 + d_+ x^+ + d_- x^- + d_2 x^+ x^-\,, \tag{71}$$

for constants $d_0$, $d_+$, $d_-$, and $d_2$. The functions $I^+$ and $I^-$ are then given by the following integrals of $T_{++}$ and $T_{--}$

$$I^\pm(x^+, x^-) = \int_{u^\pm}^{x^\pm} \mathrm{d}y\, (y - x^\mp)(y - x^\pm)\, T_{\pm\pm}(y)\,, \tag{72}$$

with each integral beginning at some point $u^+$ or $u^-$. This solution of the constraints (68) also satisfies the EOM (67) if the coefficients of the linear terms in (71) satisfy

$$d_+ - d_- + 2\, b = 0\,. \tag{73}$$

It is convenient to reparameterize the vacuum solution (71) in terms of a new set of constants that automatically obey this condition.

Switching back to the original variable $e^{2\omega} = X^{-2}\, e^{2\hat{\omega}}$, a general solution for our model with non-zero matter fields can be written in conformal gauge as

$$X = -b + \frac{c_0 + c_1\,(x^+ + x^-) + c_2\, x^+ x^- + I^+(x^+, x^-) - I^-(x^+, x^-)}{x^+ - x^-}\,, \tag{74}$$

$$\mathrm{d}s^2 = \frac{-4\,\mathrm{d}x^+\mathrm{d}x^-}{\left(c_0 + c_1\,(x^+ + x^-) + c_2\, x^+ x^- - b\,(x^+ - x^-) + I^+(x^+, x^-) - I^-(x^+, x^-)\right)^2}\,, \tag{75}$$

where $c_0$, $c_1$, and $c_2$ are arbitrary constants. With this parameterization the vacuum solutions $I^\pm = 0$ have scalar curvature given by

$$R = 2\left(c_1^2 - c_0\, c_2 - b^2\right), \tag{76}$$

which is constant, as expected.

Obtaining the general solution with matter is straightforward in Poincaré coordinates $W^\pm = x^\pm$, but throughout the rest of the paper we will also use representations of the $\hat{g}$ metric (64) with other choices of $W^\pm$. Then solutions are obtained by making the transformations $x^\pm \to W^\pm(x^\pm)$ and recalling the factor $\partial_+ W^+ \partial_- W^-$ that appears in $e^{2\hat{\omega}}$. For example, vacuum solutions can be written as

$$X = -b + \frac{c_0 + c_1\left(W^+ + W^-\right) + c_2\, W^+ W^-}{W^+ - W^-}\,, \tag{77}$$

$$ds^2 = \frac{-4\,dW^+ dW^-}{\left(c_0 + c_1\left(W^+ + W^-\right) + c_2\, W^+ W^- - b\left(W^+ - W^-\right)\right)^2}\,, \tag{78}$$

for monotonic functions $W^+(x^+)$ and $W^-(x^-)$. Let us check this result by reproducing the vacuum solutions with horizon in Schwarzschild coordinates from section 2.2. Static solutions with curvature $R = 2\left(\mu - b^2\right)$ are obtained by setting $c_0 = 1$, $c_1 = 0$, $c_2 = -\mu$,[6] and taking $W^\pm$ to be

$$W^\pm = \frac{1}{\sqrt{\mu}}\tanh\left(\frac{\sqrt{\mu}\,(t \pm z)}{2}\right)\,, \tag{79}$$

with $-\infty < t < \infty$ and $0 \le z < \infty$. In these coordinates, the dilaton and line element are

$$X = -b + \sqrt{\mu}\coth\left(\sqrt{\mu}\,z\right)\,, \tag{80}$$

$$ds^2 = \frac{-dt^2 + dz^2}{\left(\sqrt{\mu}\cosh(\sqrt{\mu}\,z) - b\sinh(\sqrt{\mu}\,z)\right)^2}\,. \tag{81}$$

Then the change of coordinate

$$z = \frac{1}{\sqrt{\mu}}\coth^{-1}\left(\frac{1 + b\,r}{r\,\sqrt{\mu}}\right)\,, \tag{82}$$

brings the solution into the Schwarzschild form (8), with $X = 1/r$ and $\xi$ given by (17).

Before constructing an explicit example of a solution with non-zero matter, it is worth taking a closer look at the change of variable used in this section and what it tells us about vacuum solutions.

## 4.3   Vacuum solutions, $X = 0$, and conformal diagrams

The change of variable (60) suggests an interesting way of thinking about these models and the properties of their solutions. As initially formulated, the model has EOM and constraints governing the conformal factor $\omega$, dilaton $X$, and matter field $f$. After the change of variable $\omega = \hat{\omega} - \ln X$ the equation for the rescaled conformal factor $\hat{\omega}$ decouples from $X$ and $f$, revealing a fixed $AdS_2$ geometry (64) hidden in the theory. The geometry described by $\omega$ then depends on $X$ and $f$, which satisfy (61) and (57) on this fixed $AdS_2$. The geometries described by $\omega$ and $\hat{\omega}$ can have qualitatively different properties, such as different conformal boundaries. In the equation describing the change of variables

$$ds^2 = X^{-2}\,d\hat{s}^2\,, \tag{83}$$

we can view the dilaton as playing the role of a defining function for the conformal boundary of the spacetime $M$ with metric $g$ [32, 33, 36, 42]. Borrowing a bit of terminology, we will refer

---

[6]The dilaton can always be brought into this form by an $SL(2, \mathbb{R})$ transformation that leaves (64) invariant. Such transformations shift the constants $c_i$ in (77) but do not change $b$ or the Ricci scalar (76).

to $g$ as the "physical" metric and $\hat{g}$ as the "unphysical" metric.[7] The interpretation of $X$ in this context as a defining function is somewhat delicate, depending on two particular properties of the solutions found in the previous section. First, $X \to \infty$ at the conformal boundary of the unphysical $AdS_2$ spacetime in precisely the right way to render $ds^2$ regular there. And second, $X = 0$ in the interior of the unphysical $AdS_2$ where $d\hat{s}^2$ is regular. Then the locus $X = 0$ describes the conformal boundary of the physical spacetime. As one would expect, this conformal boundary is timelike, null, or spacelike for $R < 0$, $R = 0$, and $R > 0$, respectively.

Let us develop this picture in more detail for the vaccum solutions, starting with the hatted $AdS_2$ written in global coordinates $(T, \sigma)$. Taking $W^{\pm} = \tan(\frac{1}{2}(T \pm \sigma))$ in (64) the line element $d\hat{s}^2$ is

$$d\hat{s}^2 = \frac{1}{\sin^2 \sigma}\left(-dT^2 + d\sigma^2\right). \tag{84}$$

The two components of the timelike conformal boundary are located at the endpoints of $0 < \sigma < \pi$, and the compactified time coordinate runs from $-\pi \le T \le \pi$. Let $\mathcal{D}$ denote the compact region $0 \le \sigma \le \pi$, $-\pi \le T \le \pi$, which includes the conformal boundary. With the matter field set to zero and the parameterization used in the previous section, the dilaton on $\mathcal{D}$ is

$$X = \frac{(1+\mu) \cos T + (1-\mu) \cos \sigma - 2b \sin \sigma}{2 \sin \sigma}. \tag{85}$$

Vacuum solutions of our theory are given by the dilaton $X$ and the physical metric $g = X^{-2} \hat{g}$ on a region in $\mathcal{D}$ where $X > 0$. The boundary of this region consists of portions of $\partial \mathcal{D}$ where the physical metric is regular, along with one or more curves $X = 0$ on the interior of $\mathcal{D}$ that describe the conformal boundary of the physical spacetime.

For finite values of $b$ and $\mu$ the condition $X > 0$ describes either a single region or two disconnected regions covering part (but not all) of $\mathcal{D}$. Several examples are shown in Figure 2. With the parameterization used here, $X > 0$ always includes a portion of $\partial \mathcal{D}$ along $\sigma = 0$ which is the $X \to \infty$ boundary of our model. It is part of the conformal boundary of the unphysical $AdS_2$ metric, but the physical metric $g = X^{-2} \hat{g}$ is regular[8] there because of the $\sin^2 \sigma$ factor in $X^{-2}$. The region $X > 0$ includes the full interval $-\pi < T < \pi$ along $\sigma = 0$ for solutions with $\mu < 0$, and a smaller interval for $\mu > 0$. For solutions with $\mu \ge 0$ the region $X > 0$ includes part of the other component of the $AdS_2$ conformal boundary at $\sigma = \pi$, where the behavior of $X$ again renders the physical metric regular. The region $X > 0$ may also include portions of the $T = \pm \pi$ components of $\partial \mathcal{D}$ where $\hat{g}$ and $X^{-2} \hat{g}$ are both regular. The rest of the boundary of the $X > 0$ region consists of one or more curves $X = 0$ on the interior of $\mathcal{D}$ described by

$$(1+\mu) \cos T + (1-\mu) \cos \sigma - 2b \sin \sigma = 0. \tag{86}$$

The $AdS_2$ metric $\hat{g}$ is regular everywhere on the interior of $\mathcal{D}$, so the locus $X = 0$ describes the conformal boundary of the physical spacetime with metric $X^{-2} \hat{g}$. A quick calculation gives

$$\frac{\partial T}{\partial \sigma} = \pm \frac{(1-\mu) \sin \sigma + 2b \cos \sigma}{\sqrt{\left((1-\mu) \sin \sigma + 2b \cos \sigma\right)^2 + 4(\mu - b^2)}}, \tag{87}$$

---

[7]Note that $X$ and $\hat{g}$ do not have quite the same properties as the defining function and unphysical metric used in, for instance, the definition of an asymptotically simple spacetime in [42]. But the construction is similar enough to justify the terminology.

[8]More generally, factors of $W^+ - W^-$ in (64) and (77) cancel in $X^{-2} \hat{g}$.

along the components of $X = 0$, which satisfies

$$\left|\frac{\partial T}{\partial \sigma}\right| \begin{cases} > 1 & \mu - b^2 < 0, \\ = 1 & \mu - b^2 = 0, \\ < 1 & \mu - b^2 > 0. \end{cases} \tag{88}$$

**Figure 2:** Examples of the region $X > 0$ in $\mathcal{D}$ for models with different values of $b$ and solutions with different values of $\mu$. The region $X > 0$ is shaded and $X = 0$ is indicated by a bold dashed black line. The parts of $\sigma = 0$ and $\sigma = \pi$ in $X > 0$, where $X \to \infty$, are indicated with a bold red line. The Schwarzschild-type coordinates introduced in section 2.2 cover the entire region $X > 0$ (or the part attached to $\sigma = 0$, in the case of two disconnected regions) for $\mu \leq b^2$ when $b \geq 0$, and for $\mu < 0$ when $b < 0$. Otherwise they cover a single horizon wedge, shaded in orange, that includes the $X \to \infty$ boundary at $\sigma = 0$. A second horizon wedge attached to $\sigma = \pi$ is not shown.

Thus, the same condition that determines the sign of the curvature as negative, zero, or positive controls whether the conformal boundary $X = 0$ is timelike, null, or spacelike, respectively.

In section 2.2 the vacuum solutions were classified according to whether or not there is a horizon at some $r_h > 0$ in Schwarzschild-type coordinates. For the horizonless solutions these coordinates cover the entire region $X > 0$, or one of the two disconnected regions when $b > 0$ and $0 < \mu < b^2$. But for solutions with horizon ($\mu > b^2$ in models with $b \geq 0$, or $\mu \geq 0$ in models with $b < 0$) they cover only part of $X > 0$. In that case $X > 0$ is a single region in $\mathcal{D}$ that includes portions of both $\sigma = 0$ and $\sigma = \pi$. There are two horizon wedges, one that includes a segment along $\sigma = 0$ and another that includes a segment along $\sigma = \pi$, which meet at the point $T = 0$, $\sigma = \cos^{-1}((\mu-1)/(\mu+1))$. The Schwarzschild-type coordinates used in previous sections cover the horizon wedge that includes the $X \to \infty$ boundary at $\sigma = 0$. We saw this explicitly in the example at the end of section 4.2, where the coordinates $t$, $z$ cover the region described in AdS$_2$ global coordinates by $-2\tan^{-1}(1/\sqrt{\mu}) < T - \sigma < T + \sigma < 2\tan^{-1}(1/\sqrt{\mu})$. This horizon patch is shown in the relevant parts of Figure 2 as the shaded orange regions.

We can construct conformal diagrams for the solutions with physical metric $g = X^{-2}\hat{g}$ as the image of the region $X > 0$ in global coordinates for dS$_2$, Minkowski$_2$, or AdS$_2$. The dilaton transforms under the isometries of the physical metric[9] so this construction is not invariant. But the causal structure of the spacetime and the important properties of the curves $X = 0$ and $X \to \infty$ are not affected, so essential features are independent of the choice of representation of $X$. Figure 3 shows a collection of conformal diagrams for different signs of $b$ and values of $\mu$. As mentioned in section 2.2, the region $X > 0$ covers only part of the full conformal diagram of AdS$_2$, Minkowski$_2$, or dS$_2$. The conformal boundary of the locally AdS$_2$ solutions without horizon has a single timelike component, rather than two, while the conformal boundary of locally AdS$_2$ solutions with horizon includes part (but not all) of both timelike components. Locally Minowski$_2$ solutions of the $b = 0$ model have either the left or right components of past and future null infinity, or part (but again not all) of both the left and right components of past and future null infinity for models with $b \neq 0$. Finally, locally dS$_2$ solutions include part but not all of the future and past components of the spacelike conformal boundary of dS$_2$. In the case of the $b = 0$ model, one can glue together two copies of the same solution along $\sigma = 0$ with the identification $\sigma \leftrightarrow -\sigma$, but in models with $b \neq 0$ this introduces a Dirac delta term in the curvature.

## 4.4 Bubble nucleation through shockwaves

As an explicit example of a solution with a non-zero matter field, consider a vacuum solution with $c_0 = 1$, $c_1 = 0$, $c_2 = -\mu$ and a matter pulse with stress-energy given approximated by

$$T_{--} = \delta\mu\,\delta(W^-), \qquad T_{++} = 0. \tag{89}$$

Then $I^+ = 0$, and the integral for $I^-$ can be evaluated to give the solution

$$X = -b + \frac{1 - (\mu + \delta\mu\,\theta(W^-))W^+W^-}{W^+ - W^-}, \tag{90}$$

$$\mathrm{d}s^2 = \frac{-4\,\mathrm{d}W^+\,\mathrm{d}W^-}{(1 - b(W^+ - W^-) - (\mu + \delta\mu\,\theta(W^-))W^+W^-)^2}. \tag{91}$$

This represents a spacetime with a matter pulse moving along the null trajectory $W^- = 0$. The curvature is $R = 2(\mu - b^2)$ before the pulse, and $R = 2(\mu + \delta\mu - b^2)$ after. If the averaged null energy condition holds then $\delta\mu \geq 0$, implying that $R$ can only increase through this process.

---

[9]These are SL$(2, \mathbb{R})$ transformations for locally (A)dS$_2$ solutions, and ISO$(1,1)$ transformations for locally Minkowski$_2$ solutions.

**Figure 3:** Conformal diagrams for the physical spacetimes with metric $g = X^{-2}\,\hat{g}$. The shaded region is a representation of $X > 0$ on the full conformal diagram of $\mathrm{dS}_2$, Minkowski$_2$, or $\mathrm{AdS}_2$. The region normally covered by the appropriate global coordinates is indicated by a purple outline. The conformal boundary $X = 0$ is shown as a bold dashed black line, while $X \to \infty$ is shown as a bold red curve. One of the two horizon wedges associated with Schwarzschild-type coordinates is shaded orange.

The analogous solution in the AP model describes an infalling pulse of matter that increases the mass of a black hole from $\mu$ to $\mu + \delta\mu$ [10, 13]. Here it represents nucleation of a region where the constant curvature has increased by $\delta R = 2\,\delta\mu$. For example, consider the $\mathrm{dS}_2$ solution of the model with $b = 0$. The solution initially describes the static patch of $\mathrm{dS}_2$ with curvature $R = 2\mu$. Using the same coordinates as the example at the end of section 4.2, the

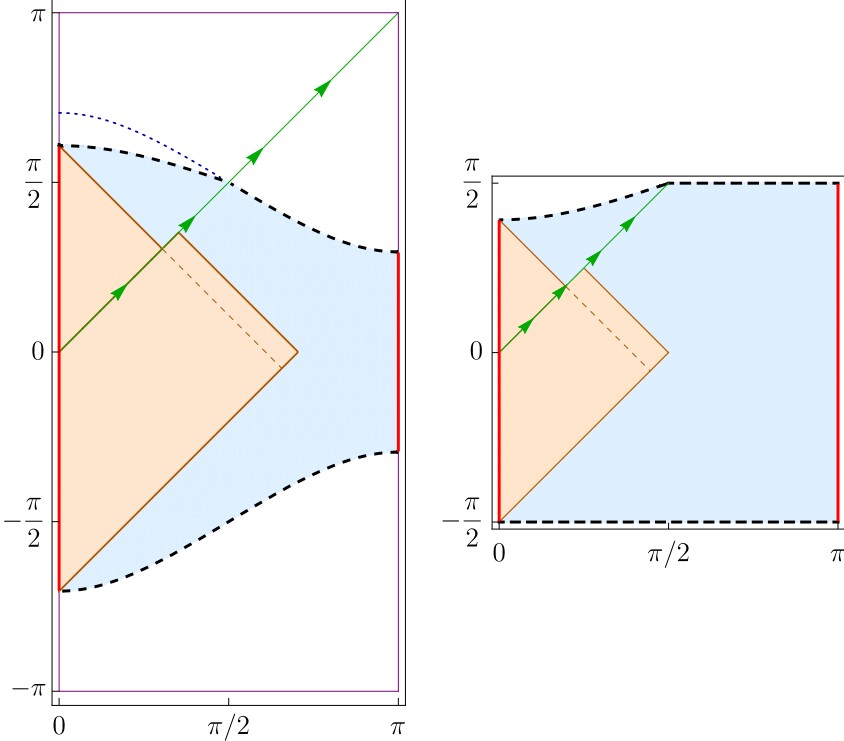

Figure 4: A solution describing a matter pulse $T_{--} = \delta\mu\,\delta(x^-)$ on a $\mu > 0$ solution of the $b = 0$ model. The diagram on the left shows the relevant regions on the conformal diagram of the unphysical AdS$_2$ spacetime, using the same conventions as Figure 2. The shaded $X > 0$ regions for vacuum solutions before ($\mu$) and after ($\mu + \delta\mu$) the pulse are joined along the green line $T - \sigma = 0$. The $X = 0$ curve for the vacuum solution $\mu$ is continued into the $T - \sigma > 0$ region as a dotted line. The diagram on the right shows the corresponding region on the (right half of the) dS$_2$ conformal diagram for the $\mu > 0$ spacetime. In both diagrams the diagonal dashed line indicates a horizon that separates part of the original static patch from an observer at $r = 0$.

pulse appears at $r = 0$ at time $t = 0$ and moves outward along the trajectory

$$r = \frac{1}{\sqrt{\mu}}\tanh\left(t\sqrt{\mu}\right). \tag{92}$$

Inside the bubble the curvature is $R = 2(\mu + \delta\mu)$. Figure 4 shows this process on both the global patch of the unphysical AdS$_2$ spacetime and the conformal diagram of the physical dS$_2$ spacetime.

As we saw in the previous section, the solution with horizon is the state of lowest free energy in the canonical ensemble and is thermodynamically stable. The analysis there set all matter fields to zero, so one might ask whether turning on matter fields introduces a potential instability via bubble nucleation. This seems unlikely. The boundary conditions of the theory fix the value $\mu = (2\pi/\beta)^2$ for the regular solution with horizon. Since bubble nucleation changes $\mu$, the spacetime inside the bubble has the "wrong" period for the ensemble's boundary conditions and exhibits a conical singularity. But such field configurations generically have larger free energy than the regular solution [43], so spontaneous bubble nucleation should be suppressed.

# 5 Semiclassical backreaction

After the investigation of the classical theory in the last few sections, we consider semiclassical effects. In the first part of this section we discuss consistent definitions of the quantum theory for the matter fields. After that, the main questions are how quantized matter field fluctuations backreact on the classical solutions, and the effect this has on the spectrum and thermodynamics. In general, the analysis in this section is valid when the quantum corrections from integrating out matter fields are subleading compared to the classical behavior of the metric and dilaton. Presumably there are other corrections that must be taken into account in regimes where semiclassical corrections are not small. Nevertheless, in the two examples we consider it is possible to solve the semiclassical EOM exactly under certain assumptions.[10] In the first example, only linearized solutions appear to be consistent. Most of the horizonless solutions of the classical theory are eliminated in that case, but the solution with horizon always remains. In the second example, the full spectrum of the classical theory is retained, and the solutions of the semiclassical EOM are consistent even when the parameter controlling corrections becomes large. The assumption is that the results are only valid at leading order, but the full results are presented in case their regime of applicability extends beyond this.

## 5.1 Matter theory and conformal anomaly

Working directly in Euclidean signature we increase the number of identical matter fields in (29) to $N$, yielding the matter action

$$\Gamma_m = \frac{1}{4\kappa^2} \sum_{i=1}^{N} \int_M d^2x \sqrt{g} \, (\partial f_i)^2 \,. \tag{93}$$

In principle one can include dilaton-dependent coupling functions in the matter action. However, such couplings lead to non-local terms in the semiclassical EOM so we consider only the minimal coupling to the metric.

The first step of the semiclassical approximation consists of fixing a specific classical background $\{g_B, X_B, f_{i,B}\}$ which we take to be a vacuum solution. Next, we include quantum effects from fluctuations of the matter fields $\delta f_i$, integrating over these configurations in the path integral while treating the metric and dilaton in the saddle-point approximation. Evaluating the path integral requires a specific choice of measure for the matter fields, which may depend on the dilaton in addition to the metric. Here, two crucial ingredients enter: boundary conditions are needed for the matter fluctuations, and we need to require $G^{-1} \gg N$ for keeping the perturbations around the gravitational saddle-point small when backreaction is taken into account. We define the parameter

$$\Delta := \frac{NG}{3} \,, \tag{94}$$

so that $\Delta \ll 1$ when $G^{-1} \gg N$.

Compared to the models studied in [10, 20] we allow a more general class of measures for the matter path integral. They are defined by

$$1 = \int \mathcal{D} \delta f_i \, e^{-\|\delta f_i\|^2} \,, \quad \forall i \in \{1, \dots, N\} \,, \tag{95}$$

where

$$\|\delta f_i\|^2 := \int_M d^2x \sqrt{g_B} \, X_B^{\eta} \, \delta f_i^2 \,, \quad \eta \in \mathbb{R} \,. \tag{96}$$

---

[10]When these assumptions are relaxed a perturbative treatment is necessary, discussed in Appendix B.

Note that this choice preserves diffeomorphism invariance but breaks the Weyl invariance of the classical theory. Although $\eta \neq 0$ might seem unusual we take the point of view that from a 2D perspective it is as justified as any other choice, provided the measure is well-defined.[11] To check this we need to make sure that (96), which is the exponent in (95), is finite for field configurations allowed by the boundary conditions for the scalar fields. We fix these boundary conditions by considering static solutions that approach the vacuum $f_{i,B} = 0$ in the asymptotic region, i.e. at $X_B \to 0$ or $r \to 0$. To determine the fall-off behavior consider solutions to the massless Klein–Gordon equation

$$\nabla_\mu \left( g_B^{\mu\nu} \partial_\nu f_i \right) = 0, \tag{97}$$

where the covariant derivative is compatible with the background metric. Imposing staticity in the matter sector, $f_i = f_i(r)$, and taking (13),(17) as the background we have

$$\partial_r \left( \xi_B(r) \partial_r f_i(r) \right) = 0, \tag{98}$$

which has the general solution

$$f_i(r) = c_0 + c_1 \int_0^r \mathrm{d}\rho \, \frac{1}{\xi_B(\rho)}, \quad c_0, c_1 \in \mathbb{R}. \tag{99}$$

The integration constants are chosen to be the same for all of the fields as we want to treat them identically. This solution approaches the vacuum configuration $f_{i,B} = 0$ at $r \to 0$ if $c_0 = 0$. Then

$$\lim_{r \to 0} f_i(r) = c_1 \, r - b \, c_1 \, r^2 + \mathcal{O}(r^3), \tag{100}$$

and we conclude that fluctuations of the matter field behave near the boundary as

$$\lim_{r \to 0} \delta f_i(r) = \mathcal{O}(r). \tag{101}$$

In that case the contribution to the normalization integral (96) in a neighborhood of $r = 0$ is finite if

$$\eta < 3. \tag{102}$$

This places an upper bound on the possible measures we can define.

Integrating out the matter fields is straightforward as we are dealing with a Gaussian integral. This gives a non-local effective action $W[g,X]$ for the metric and dilaton, with one-loop expectation values for the source terms in their EOM given by

$$\langle T^{\mu\nu} \rangle = \frac{2}{\sqrt{g}} \frac{\delta W}{\delta g_{\mu\nu}}, \quad \langle T_X \rangle = \frac{2}{\sqrt{g}} \frac{\delta W}{\delta X}. \tag{103}$$

For general $\eta$ subject to the condition (102) the effective action is of generalized Polyakov type. Without knowing the full details of this action we can still draw a few important conclusions. First, diffeomorphism invariance has been preserved during quantization, so the expectation values (103) satisfy the generalized conservation equation [6]

$$\nabla^\mu \langle T_{\mu\nu} \rangle = \frac{1}{2} \langle T_X \rangle \, \partial_\nu X. \tag{104}$$

---

[11]Choices of $\eta \neq 0$ can also arise for dilaton models derived from higher-dimensional theories such as spherically reduced Einstein gravity. There, one gets $\eta = 1$ with an additional dilaton dependent coupling in the matter action [6].

Second, breaking Weyl invariance causes the trace of the one-loop stress tensor to acquire a non-zero value. Its form can be obtained via heat kernel methods [6, 44] and is given by

$$\langle T^\mu{}_\mu \rangle = \frac{N}{24\pi}\Big(R + \eta\,\frac{(\nabla X)^2}{X^2} - \eta\,\frac{\nabla^2 X}{X}\Big). \tag{105}$$

In conformal gauge (9) this fixes the component $\langle T_{+-}\rangle$

$$\langle T_{+-}\rangle = -\frac{1}{4}\,e^{2\omega}\,\langle T^\mu{}_\mu\rangle = \frac{N}{24\pi}\Big(-2\,\partial_+\partial_-\omega - \eta\,\partial_+\partial_-\ln X\Big). \tag{106}$$

Our goal is to determine the semiclassical backreaction on the geometry and dilaton, for which we need the remaining components of the one-loop source terms. These are determined, up to various ambiguities, by solving the generalized conservation equation (104). In conformal gauge, its components are

$$\partial_+\langle T_{+-}\rangle - 2\partial_+\omega\langle T_{+-}\rangle = -\partial_-\langle T_{++}\rangle - \frac{1}{4}e^{2\omega}\langle T_X\rangle\partial_+ X\,, \tag{107}$$

$$\partial_-\langle T_{+-}\rangle - 2\partial_-\omega\langle T_{+-}\rangle = -\partial_+\langle T_{--}\rangle - \frac{1}{4}e^{2\omega}\langle T_X\rangle\partial_- X\,. \tag{108}$$

They can be solved by inserting (106) on the left-hand side and bringing the resulting terms into the forms $\partial_\pm(\dots)$ or $(\dots)\partial_\pm X$.[12] Comparing this with the right-hand side determines $\langle T_{\pm\pm}\rangle$ and $\langle T_X\rangle$ up to integration functions and various ambiguities. Then the following local expressions are obtained

$$\langle T_{\pm\pm}\rangle = \frac{N}{24\pi}\Big(2\partial_\pm^2\omega - 2(\partial_\pm\omega)^2 + \eta\,\partial_\pm^2\ln X$$
$$- \gamma\,(\partial_\pm\ln X)^2 - 2\eta\,\partial_\pm\omega\,\partial_\pm\ln X + (\partial_\pm k(X))^2\Big) + \tau_{\pm\pm}(x^\pm), \tag{109}$$

$$\langle T_X\rangle = \frac{N}{3\pi}e^{-2\omega}\frac{1}{X}\Big(\eta\,\partial_+\partial_-\omega + \gamma\,\partial_+\partial_-\ln X - X\,\partial_X k(X)\,\partial_+\partial_- k(X)\Big). \tag{110}$$

These expressions depend on a free parameter $\gamma \in \mathbb{R}$ because there are terms that can be associated with either $\langle T_{\pm\pm}\rangle$ or $\langle T_X\rangle$. In addition, it is always possible to add and subtract a term of the form $(\partial_\pm k(X))^2$ to the brackets in $\langle T_{\pm\pm}\rangle$, where $k(X)$ is an arbitrary function of the dilaton. One of the two can then be rewritten as a contribution to $\langle T_X\rangle$. Finally, the $\pm\pm$ components depend on two arbitrary integration functions that we denote $\tau_{\pm\pm}(x^\pm)$. It is instructive to make the change of variable $\omega = \hat\omega - \ln X$ in the expression for $\langle T_{\pm\pm}\rangle$. Using (64), it can then be written as

$$\langle T_{\pm\pm}\rangle = \frac{N}{24\pi}\big\{W^\pm(x^\pm), x^\pm\big\} + \tau_{\pm\pm}(x^\pm) + \frac{N}{24\pi}\Big(\eta - \gamma + (X\,\partial_X k)^2\Big)(\partial_\pm\ln X)^2. \tag{111}$$

The first term is the Schwarzian derivative of $W^\pm(x^\pm)$ with respect to $x^\pm$. It is universal in the sense that it appears in $\langle T_{\pm\pm}\rangle$ for any choice of $\eta$, $\gamma$, and $k(X)$.

It is evident that at this stage there is quite some freedom left in the definition of the semiclassical theory: We can still choose a measure for the path integral and even then the one-loop effects of the matter fields are not fully determined. Part of this ambiguity corresponds to the usual integration functions $\tau_{\pm\pm}$ but there is also the freedom associated with $\gamma$ and $k(X)$. Some of these ambiguities are fixed by demanding certain properties for the effective

---

[12]At this point non-local terms would appear for a non-minimal dilaton coupling in the matter action. For instance, a matter Lagrangian of the form $X^\zeta(\partial f_i)^2$ introduces a term proportional to $X^{-2}\partial_+^2 X\partial_- X$ in (107). This term cannot be brought into one of the two forms on the right-hand side, resulting in a non-local term in either $\langle T_{\pm\pm}\rangle$ or $\langle T_X\rangle$.

action itself. In Appendix C we construct effective actions that reproduce the one-loop source terms for the metric and dilaton and discuss various properties they should have. Once these ambiguities are fixed, either by consistency or assumption, we can compute the semiclassical backreaction on the geometry and dilaton. From there, it is possible to calculate the on-shell Euclidean action and corrections to the thermodynamics. In the following sections we carry this out for two different choices of $\eta$, starting with the simplest choice, $\eta = 0$.

## 5.2 Standard semiclassical theory

As a first attempt to define a semiclassical theory we consider the simple choice $\eta = 0$ to define the measure (95)-(96). The matter path integral in this case is independent of the dilaton, so $\langle T_X \rangle = 0$ and the trace anomaly is

$$\langle T^{\mu}{}_{\mu}\rangle = \frac{N}{24\pi}R\,, \tag{112}$$

which in conformal gauge reads

$$\langle T_{+-}\rangle = -\frac{N}{12\pi}\partial_+\partial_-\omega\,. \tag{113}$$

The generalized conservation equation (104) reduces to the standard result $\nabla^{\mu}\langle T_{\mu\nu}\rangle = 0$. Solving for the flux components $\langle T_{\pm\pm}\rangle$ yields

$$\langle T_{\pm\pm}\rangle = \frac{N}{24\pi}\Big(2\,\partial_{\pm}^2\omega - 2(\partial_{\pm}\omega)^2\Big) + \tau_{\pm\pm}(x^{\pm})\,. \tag{114}$$

Since there is no dilaton-dependence in the measure it is natural to ignore the ambiguities associated with $\gamma$ and $k(X)$ in (109). The only freedom is the choice of integration functions $\tau_{\pm\pm}(x^{\pm})$.

The components $\langle T_{\mu\nu}\rangle$ act as sources in the EOM. In the following it is useful to work in Schwarzschild gauge (21). Then, the backreacted constraint and EOM read

$$0 = X^2\,\xi^2\,\partial_r\Big(\frac{1}{X^2}\,\partial_r X\Big) + \kappa^2\,\langle T_{\pm\pm}\rangle\,, \tag{115}$$

$$0 = \partial_r\big(\xi\,\partial_r X\big) - 2X^2(X+b) + \Delta\,\partial_r^2\xi\,, \tag{116}$$

$$0 = \partial_r^2\xi + 2\,\partial_r\Big(\xi\,\frac{1}{X}\,\partial_r X\Big) - 2X^2 - 2\Delta\,\frac{1}{X}\,\partial_r^2\xi\,, \tag{117}$$

with $\Delta$ defined in (94). Our goal is to solve these equations to understand semiclassical corrections to the static solutions considered in the previous sections. This will reveal certain limitations of the model with $\eta = 0$. Specifically, most of the static solutions that appeared in the classical theory turn out to be no longer present in the semiclassical theory.

Inspired by [10], we first assume that $\langle T_{\pm\pm}\rangle = 0$, which is accomplished by taking $\tau_{\pm\pm}(x^{\pm})$ proportional to the Schwarzian of $W^{\pm}(x^{\pm})$ in (111).[13] Then the constraint (115) can be integrated to give

$$X = \frac{1}{d_0\,r + d_1}\,, \tag{118}$$

for arbitrary constants $d_0$ and $d_1$. If we fix $r = 0$ as the point where $X \to +\infty$ then $d_1 = 0$ and $d_0 > 0$. Without loss of generality, we set $d_0 = 1$ (the coordinate $r$ is dimensionless). With $X = 1/r$ the EOM (116) and (117) become

$$0 = r\,\partial_r\xi - 2\xi + 2\big(1 + b\,r\big) - \Delta\,r^3\,\partial_r^2\xi\,, \tag{119}$$

$$0 = r^2\big(1 - 2r\,\Delta\big)\partial_r^2\xi - 2r\,\partial_r\xi + 2\big(\xi - 1\big)\,. \tag{120}$$

---

[13]The condition $\langle T_{\pm\pm}\rangle = 0$ is relaxed in Appendix B, where the equations are solved perturbatively.

Solving (120) gives the Killing norm

$$\xi = 1 + \frac{c_1 r + c_2 r^2}{1 - 2 r \Delta},\tag{121}$$

with integration constants $c_1$ and $c_2$. Thus, even semiclassically metric and dilaton remain stationary. Inserting this result into (119) obtains

$$0 = \frac{(2 b - c_1) r \left(1 - 6 r \Delta + 12 r^2 \Delta^2\right) - 4 r^4 \Delta^2 \left(4 b \Delta + c_2\right)}{(1 - 2 r \Delta)^3}.\tag{122}$$

We solve these equations to linear order in the backreaction parameter $\Delta$.

### 5.2.1 Linearized solution

Linearizing the backreaction depends on the type of classical solution we are considering. First, consider a solution with a horizon with $r$ taking values in the finite range $0 \le r \le r_h$. Expanding (122) for $\Delta \ll 1$ gives an expansion in powers of $r \Delta$

$$0 = r (2 b - c_1) + \mathcal{O}(r^2 \Delta^2).\tag{123}$$

As long as $r \Delta \ll 1$, which is guaranteed when $\Delta \ll 1$ and $r$ takes values in a finite range, the expansion is under control. The only condition on our solution at leading order is $c_1 = 2 b$. Rewriting the other integration constant as $c_2 = -\mu + b^2 - \Delta 4b$ yields

$$\xi = (1 + b r)^2 - \mu r^2 - \Delta R_{\scriptscriptstyle B} r^3,\tag{124}$$

which is just the classical solution (17) with a semiclassical correction proportional to the classical curvature $R_{\scriptscriptstyle B} = 2 (\mu - b^2)$.

However, we encounter a qualitatively different result for backreaction on the continuum of AdS states with $0 \le r < \infty$. In that case, the fact that $r$ becomes arbitrarily large means that $r \Delta$ will eventually become of $\mathcal{O}(1)$ for any finite $\Delta$, no matter how small. This is apparent from the coefficient of the $\partial_r^2 \xi$ term in (120). At some point the part involving $r \Delta$ becomes dominant and backreaction has a significant impact on the solution, manifesting as a potential pole in (121). The only way to avoid this pole is to tune $c_2 = -2 \Delta c_1$. Then, $\xi = 1 + c_1 r$, and (119) fixes $c_1 = 2 b$ to give

$$\xi = 1 + 2 b r.\tag{125}$$

This horizonless solution remains in the spectrum of $b \ge 0$ theories since it corresponds to the state $\mu = b^2$ with classical curvature $R_{\scriptscriptstyle B} = 0$. In that case the trace anomaly $\langle T^\mu{}_\mu \rangle$ vanishes and there are no semiclassical corrections to the EOM. So while it is possible to linearize the backreaction on solutions with a horizon when $\Delta \ll 1$, any $\Delta \ne 0$ eliminates the continuum of horizonless AdS states. The only such state that remains is the $R = 0$ solution (125) in models with $b \ge 0$.

In conclusion, the model with $\eta = 0$ does not retain the full spectrum of classical solutions once semiclassical corrections are taken into account; only solutions with horizon remain. This conclusion was reached assuming $\langle T_{\pm\pm} \rangle = 0$. Appendix B analyzes the semiclassical corrections without making this assumption, reaching essentially the same conclusions, i.e., the model defined with $\eta = 0$ in the measure does not retain the full spectrum of the classical theory. As discussed around (206), the only way to avoid this is to define the matter path integral with $\eta = 2$. This leads to an an exactly solvable model with more or less the same spectrum as the classical theory, analyzed in section 5.3.

### 5.2.2 Semiclassical thermodynamics

Before turning to the exactly solvable $\eta = 2$ model, it is worth considering the semiclassical thermodynamics for $\eta = 0$.

Recall that for a given value of $b$ a classical solution with horizon exists only for a specific range of $\beta$ (see table 1). There is the additional condition $r_h \Delta \ll 1$ which can further restrict the allowed values of $\beta$. To first order in $\Delta$, the semiclassical solution (124) has a horizon at

$$r_h = \frac{1}{\sqrt{\mu} - b}\left(1 - \Delta\,\frac{\sqrt{\mu} + b}{\sqrt{\mu}(\sqrt{\mu} - b)}\right). \tag{126}$$

Regularity at this horizon requires

$$\beta = -\frac{4\pi}{\partial_r \xi}\bigg|_{r_h}, \tag{127}$$

which determines a specific value of $\mu$. The location of the horizon can then be expressed in terms of $\beta$ as

$$r_h = \frac{\beta}{2\pi - b\beta}\left(1 + \Delta\,\frac{\beta(2\pi + b\beta)}{(2\pi - b\beta)^2}\right). \tag{128}$$

For models with $b > 0$, the condition $\Delta r_h \ll 1$ breaks down near the upper end of the classically allowed range $0 < \beta < 2\pi/b$, where $r_h$ is unbounded. Instead, in these models the linearized solution is only consistent when $\beta \ll 2\pi/b$ and $\Delta \ll b$. This corresponds to the high temperature limit $T \gg b/2\pi$ of the classical theory. For $b = 0$ the classically allowed range $0 < \beta < \infty$ becomes $0 < \beta \ll 2\pi/\Delta$, with $\Delta \ll 1$. Finally, in models with $b < 0$, the factors of $(2\pi + |b|\beta)^{-1}$ in $r_h$ are bounded and $\Delta r_h \ll 1$ for all $0 < \beta < \infty$ when $\Delta \ll |b|$. This is the same range of $\beta$ that is allowed in the classical theory. However, there is an additional condition that we shall encounter momentarily which places a lower limit on the allowed temperature when $b < 0$.

To study the thermodynamics of this model with semiclassical corrections we work with the local form of the effective action introduced in Appendix C,

$$\begin{aligned}
\Gamma_{\text{eff}} = &-\frac{1}{2\kappa^2}\int_M d^2x\,\sqrt{g}\left(X R + \frac{2}{X}(\nabla X)^2 + 2X^3 + 2bX^2\right)\\
&-\frac{1}{\kappa^2}\int_\Sigma dx\,\sqrt{h}\left(X K - \sqrt{X^4 + 2bX^3}\right)\\
&+\frac{\Delta}{\kappa^2}\int_M d^2x\,\sqrt{g}\left(\chi R + (\nabla\chi)^2\right) + \frac{2\Delta}{\kappa^2}\int_\Sigma dx\,\sqrt{h}\,\chi K.
\end{aligned} \tag{129}$$

The EOM for the ancillary scalar field is

$$2\nabla^2\chi = R. \tag{130}$$

Working in conformal gauge the solution on the static background is

$$\chi = -\omega - \sqrt{\mu}\,z + \chi_0. \tag{131}$$

There is an arbitrary constant $\chi_0$ in the homogeneous solution along with a linear term whose coefficient is fixed by requiring finite $\chi$ at the horizon. In the Schwarzschild coordinates (21), where $X = 1/r$, this reads

$$\chi = \chi_0 - \ln 2 - \coth^{-1}\left(\frac{1 + br}{r\sqrt{\mu}}\right) - \frac{1}{2}\ln\left((1 + br)^2 - \mu r^2\right). \tag{132}$$

The effective Newton's constant in (129) is $G_{\text{eff}}^{-1} = G^{-1}(X - 2\Delta\chi)$, and we expect that this should remain non-negative. The factor of $X - 2\Delta\chi$ is a monotonically decreasing function of $r$ that takes its smallest value at the horizon (128), where the dilaton and ancillary scalar are given by

$$X_h = \frac{1}{r_h} = 2\pi T - b + \Delta\frac{2\pi T + b}{2\pi T - b}, \tag{133}$$

$$\chi_h = \chi_0 + \ln\frac{2\pi T - b}{8\pi T}. \tag{134}$$

In models with $b > 0$ the condition $r_h\Delta \ll 1$ excludes temperatures $T \sim b/2\pi$ near the poles in $X_h$ and $\chi_h$. But for $b < 0$ models, where $r_h\Delta \ll 1$ is consistent for all $T \geq 0$, the ln term in $\chi_h$ becomes large when $T$ is very small. Requiring $G_{\text{eff}}^{-1} > 0$ places a lower bound on the temperature in models with $b < 0$, given by

$$T > \frac{|b|}{8\pi}\exp\left(-\frac{|b|}{2\Delta}\right). \tag{135}$$

Since $\Delta \ll |b|$ this is exponentially small compared to $|b|/2\pi$.

As in the classical theory, the canonical ensemble for the semiclassical theory is dominated by the solution with horizon. In models with $b > 0$ this solution is only consistent in the high temperature limit $T \gg b/2\pi$, and for $b = 0$ the allowed temperatures are $T \gg \Delta/2\pi$. In both cases the solution with horizon is locally $\text{dS}_2$. But for $b < 0$ we again have distinct temperature regimes. At temperatures $T > |b|/2\pi$ the horizon solution that dominates the canonical ensemble is locally $\text{dS}_2$, and at $T = |b|/2\pi$ it is locally Minkowski. For temperatures below $|b|/2\pi$ the horizon solution is locally $\text{AdS}_2$. The temperature range for locally $\text{AdS}_2$ solutions is

$$\frac{|b|}{8\pi}\exp\left(-\frac{|b|}{2\Delta}\right) < T < \frac{|b|}{2\pi}. \tag{136}$$

Since the lower bound is exponentially small it is possible to consider a low-temperature limit $T \ll |b|/2\pi$. In all cases, the solution with horizon is stable since its free energy is negative and there is no longer a continuum of competing saddle points to consider. Evaluating the on-shell Euclidean action (129) to first order in $\Delta$, and restoring factors of $a$ that were previously set to 1, the free energy is

$$F = -\frac{1}{2\kappa^2 a}(2\pi T - a b)^2 + \frac{\Delta}{\kappa^2}(a b + 2\pi T + 4\pi T \chi_h). \tag{137}$$

The horizon value of the ancillary scalar, which depends on $T$ when $b \neq 0$, is given in (134). The entropy that follows from this has the expected form

$$S = -\frac{\partial F}{\partial T} = \frac{2\pi}{\kappa^2}(X_h - 2\Delta\chi_h), \tag{138}$$

and the internal energy is

$$E = F + T S = \frac{1}{2\kappa^2 a}(2\pi T - a b)(2\pi T + a b) - \frac{\Delta}{\kappa^2}a b\frac{2\pi T + a b}{2\pi T - a b}. \tag{139}$$

Let us consider these results for models with different signs of $b$.

For $b > 0$ the apparent pole in the semiclassical correction to $E$ is always outside of the allowed range of $T$. In those models the conditions $T \gg b/2\pi$ and $b \gg \Delta$ ensure that the leading semiclassical corrections are much smaller than the sub-leading classical terms. In models with $b = 0$ the semiclassical correction to $E$ vanishes entirely. It is interesting to note

that $\chi_h$ is independent of $T$ in that case, and hence can be set to zero with the choice $\chi_0 = \ln 4$. Then the $b = 0$ model has

$$F = -\frac{2\pi T^2}{a\,\kappa^2} + \frac{\Delta}{\kappa^2}\,2\pi T\,, \qquad S = \frac{2\pi}{\kappa^2}\,X_h = \frac{2\pi}{\kappa^2}\left(2\pi\,T - \Delta\right), \qquad E = \frac{2\pi T^2}{a\,\kappa^2}\,. \qquad (140)$$

There is no contribution from the ancillary scalar in these expressions; the order $\Delta$ terms are due to the correction to the dilaton at the horizon.

Models with $b < 0$ are especially interesting. Starting at low temperatures, the ground state transitions from a locally $AdS_2$ solution to a locally $dS_2$ solution as $T$ increases. This feature appeared in the classical theory, and persists in both this model and the exactly solvable model of the next section. For boundary conditions $T = |b|/2\pi$ the classical solution is locally Minkowski. In that case $\langle T^\mu{}_\mu \rangle = 0$, the semiclassical corrections vanish, and it makes sense to fix the constant part of the ancillary field to $\chi_0 = \ln 2$ so that $F$ and $S$ take the same values as in the classical theory. Below this temperature we are in a regime where the canonical ensemble is dominated by a locally $AdS_2$ solution. It is of interest then to compare the results above with the AP model. In our conventions, their model corresponds to $b = -1$ and units where $a = 2$. For $T \ll a|b|/2\pi = 1/\pi$, which is still well above the exponentially small temperature where this semiclassical approximation breaks down, we obtain

$$S = \frac{\pi T}{4\,G} + \frac{1}{4\,G} + \frac{N}{6}\,\ln T + \frac{N}{12} + \frac{N}{6}\left(\ln 4\pi - \chi_0\right), \qquad (141)$$

$$E = \frac{\pi T^2}{8\,G} - \frac{1}{8\pi\,G} - \frac{N}{12\pi} + \frac{N}{6}\,T\,. \qquad (142)$$

This differs from the results of [10] by state-independent shifts only,

$$S \simeq S_{\text{AP}} - \frac{N}{6}\,\chi_0\,, \qquad E \simeq E_{\text{AP}} - \frac{1 + 2\Delta}{8\pi\,G}\,. \qquad (143)$$

At low temperatures the behavior of the two models is essentially the same, even though the semiclassical corrections have a different form. The difference in the entropy is proportional to the constant term in the ancillary field. This is set to zero in [10], but as noted above there may be more natural choices. The internal energy is independent of $\chi_0$ and agrees with the Brown–York quasilocal stress tensor obtained from (129). The difference between $E$ and $E_{\text{AP}}$ in (143) is a constant term which is already present in the classical result and receives a semiclassical correction. Though small, the correction could be comparable to the $T^2$ part of the classical result in this low temperature limit.

## 5.3   Exactly solvable semiclassical theory

We turn to the model with $\eta = 2$. In that case, the factor of $\sqrt{g_{_B}}X_{_B}^2$ in (96) is precisely $\sqrt{\hat{g}_{_B}}$, and the measure for the matter fields is the same one used in [10]. Since the definition of the measure for the matter fields depends on $X$ there are additional ambiguities in how $\langle T_X \rangle$ and the components of $\langle T_{\mu\nu} \rangle$ are defined. Based on the analysis of Appendix B, the only consistent possibility for the ambiguity $k(X)$ in (109) and (110) can be absorbed in a redefinition of $\gamma$, so henceforth we set $k(X) = 0$. This leaves the parameter $\gamma$ as a remaining ambiguity in the definition of the model. In appendix C we construct a family of effective actions consistent with the Weyl anomaly in this model. Demanding that these actions all take the same value for a given solution of the semiclassical EOM fixes $\gamma = 2$, and also identifies a unique homogeneous solution for the ancillary scalar $\chi$ for horizonless solutions.

### 5.3.1 Effective action and semiclassical equations of motion

A convenient representative of the equivalent effective actions derived in appendix C is

$$
\begin{aligned}
\Gamma_{\text{eff}} = &-\frac{1}{2\kappa^2} \int_M d^2x \sqrt{g} \left( X R + \frac{2}{X} (\nabla X)^2 + 2X^3 + 2bX^2 \right) \\
&+ \frac{\Delta}{\kappa^2} \int_M d^2x \sqrt{g} \left( \chi R + (\nabla\chi)^2 + 2\nabla\chi\nabla\ln X \right) \\
&- \frac{1}{\kappa^2} \int_\Sigma dx \sqrt{h} \left( X K - \sqrt{X^4 + 2bX^3} \right) + \frac{2\Delta}{\kappa^2} \int_\Sigma dx \sqrt{h} \left( \chi K + \frac{X}{2} \right).
\end{aligned}
\tag{144}
$$

The EOM for the ancillary scalar $\chi$ is

$$
2\nabla^2\chi = R - 2\nabla^2\ln X.
\tag{145}
$$

Variations of the action (144), evaluated on solutions of (145), reproduce the components of the Weyl anomaly with $\eta = 2$, $\gamma = 2$, and $k(X) = 0$

$$
\langle T_{+-} \rangle = -\frac{N}{24\pi} \partial_+\partial_- \left( 2\omega + 2\ln X \right) = -\frac{N}{12\pi} \partial_+\partial_-\hat{\omega},
\tag{146a}
$$

$$
\langle T_X \rangle = \frac{N}{3\pi} e^{-2\omega} \frac{1}{X} \partial_+\partial_- \left( 2\omega + 2\ln X \right) = \frac{2N}{3\pi} e^{-2\hat{\omega}} X \partial_+\partial_-\hat{\omega},
\tag{146b}
$$

$$
\langle T_{\pm\pm} \rangle = \frac{N}{24\pi} \left( 2\partial_\pm^2\hat{\omega} - 2(\partial_\pm\hat{\omega})^2 \right) + \tau_{\pm\pm}(x^\pm).
\tag{146c}
$$

Here, we also write the components in terms of the variable $\hat{\omega}$ defined in (60). Note that all the parts of the action involving the ancillary field are proportional to the number of matter fields $N$. Treating this as a classical problem requires $N \gg 1$ and $G \ll 1$ such that loop corrections can be neglected. Also, as initially discussed, depending on our expectation concerning additional corrections to the semiclassical EOM we may need to demand $G^{-1} \gg N$, i.e. $\Delta \ll 1$. While in principle we could linearize in $\Delta$ at any stage, the EOM can be solved exactly and we retain the full $\Delta$-dependence throughout this section. As before, the effective inverse Newton constant in (144) is

$$
\frac{1}{G_{\text{eff}}} = \frac{1}{G} \left( X - 2\Delta\chi \right),
\tag{147}
$$

and needs to be positive.

The semiclassical EOM expressed in terms of $X$ and $\hat{\omega}$ read

$$
0 = e^{2\hat{\omega}} \partial_\pm \left( e^{-2\hat{\omega}} \partial_\pm X \right) + \kappa^2 \langle T_{\pm\pm} \rangle,
\tag{148a}
$$

$$
0 = -\partial_+\partial_- X - \frac{1}{2} e^{2\hat{\omega}} (X + b) - \frac{N\kappa^2}{12\pi} \partial_+\partial_-\hat{\omega},
\tag{148b}
$$

$$
0 = \partial_+\partial_-\hat{\omega} + \frac{1}{4} e^{2\hat{\omega}}.
\tag{148c}
$$

The equation for $\hat{\omega}$ is a linear combination of the dilaton- and $g_{+-}$ EOM. Both of these equations have semiclassical source terms, but the combination of $\langle T_{+-} \rangle$ and $\langle T_X \rangle$ cancels in the equation for $\hat{\omega}$. As a result, the hatted geometry does not receive corrections and is still AdS$_2$ with curvature $\hat{R} = -2$. Equation (148b) simplifies to

$$
-\partial_+\partial_- X - \frac{1}{2} e^{2\hat{\omega}} (X + b - \Delta) = 0.
\tag{149}
$$

This is just the classical equation with $b$ shifted by $-\Delta$. Splitting the dilaton into parts as

$$
X = X_B + \Delta + Y,
\tag{150}
$$

where $X_B$ is the classical solution, the field $Y$ satisfies

$$0 = -\partial_+\partial_- Y - \frac{1}{2} e^{2\hat{\omega}} Y. \tag{151}$$

But solutions of this equation are the same as the homogeneous solutions of the equation for the classical dilaton $X_B$, and these have already been accounted for since $X$ satisfies the same boundary conditions we initially considered for $X_B$. We conclude that the dilaton with backreaction is just $X = X_B + \Delta$. The constraints (148a) then force $\langle T_{\pm\pm}\rangle$ to vanish, which is once again accomplished by taking $\tau_{\pm\pm}(x^\pm)$ proportional to the Schwarzian of $W^\pm(x^\pm)$ in (146c).

The full effect of integrating out the matter fields appears as the shift

$$b \to b - \Delta, \tag{152}$$

in the dilaton and metric (vacuum) solutions of the classical theory. There are no semiclassical corrections to the hatted geometry, so the response of the physical metric $g = X^{-2}\hat{g}$ is due entirely to its dependence on the dilaton. Unlike the model in the previous section, all solutions of the classical theory remain once semiclassical corrections are taken into account. The results for $X$ and $\hat{\omega}$ are the same as in [10].

### 5.3.2 Solutions to semiclassical equations of motion

Now we construct solutions with backreaction, evaluate the (non-linearized) on-shell action, and analyze the thermodynamics for this model. While the following analysis can in principle be carried out for general $b$, we focus on models with $b < 0$. These have the same interesting feature as the classical ones: a stable AdS$_2$ phase at low temperatures that transitions to a stable dS$_2$ phase at high temperatures.[14] Classically, they have a single regular solution with horizon for any value $\beta > 0$ of the boundary condition, along with a continuum of horizonless states (cf. table 1). The exact solution of (148) with a horizon takes the same form as the classical result, with the shift $|b| \to |b| + \Delta$. In conformal gauge we have

$$\hat{\omega}(z) = \hat{\omega}_B = \frac{1}{2}\ln\frac{4\mu}{\sinh^2(\sqrt{\mu}z)}, \tag{153a}$$

$$X(z) = |b| + \Delta + \sqrt{\mu}\coth\left(\sqrt{\mu}z\right), \tag{153b}$$

$$\omega(z) = \hat{\omega} - \ln X = \frac{1}{2}\ln\frac{4\mu}{\left(\sqrt{\mu}\cosh(\sqrt{\mu}z) + (|b|+\Delta)\sinh(\sqrt{\mu}z)\right)^2}, \tag{153c}$$

$$\chi(z) = -\hat{\omega} - \sqrt{\mu}z + \chi_0, \tag{153d}$$

where $z \in [0, \infty)$. The Ricci scalar remains constant but is shifted

$$R = 2\left(\mu - (|b|+\Delta)^2\right), \tag{154}$$

while regularity fixes $\mu$ as in (18). The ancillary field $\chi$ contains a single unfixed constant $\chi_0$ in its homogeneous solution. One might try to fix this constant as in section 5.2 by looking for a state for which the corrections should vanish. Here, however, the anomaly takes a different form with both (146a) and (146b) sourcing the EOM. A short calculation shows that the latter cannot vanish for any solution with horizon.

The horizonless states correspond to $\lambda > b^2$ in (27). However, in the following it is convenient to adopt a slightly different parameterization. Defining the non-negative parameter

$$\tilde{\lambda} = \lambda - b^2, \tag{155}$$

---

[14]We regard $\Delta$ as a small parameter compared to $b \neq 0$, so the shift (152) does not affect, for instance, the classification of different possible solutions according to the sign of $b \to b - \Delta$.

the Killing norm $\xi$ in Schwarzschild coordinates takes the form

$$\xi = \left(1 - \left(|b| + \Delta\right)r\right)^2 + \tilde{\lambda}\, r^2, \quad \tilde{\lambda} > 0. \tag{156}$$

In conformal gauge the solution is

$$\hat{\omega}(z) = \frac{1}{2}\ln\left(\frac{4\tilde{\lambda}}{\sin^2\left(\sqrt{\tilde{\lambda}}\,z\right)}\right), \tag{157a}$$

$$X(z) = |b| + \Delta + \sqrt{\tilde{\lambda}}\cot\left(\sqrt{\tilde{\lambda}}\,z\right), \tag{157b}$$

$$\omega(z) = \hat{\omega} - \ln X = \ln\left(\frac{2\sqrt{\tilde{\lambda}}}{\sqrt{\tilde{\lambda}}\cos\left(\sqrt{\tilde{\lambda}}\,z\right) + (|b| + \Delta)\sin\left(\sqrt{\tilde{\lambda}}\,z\right)}\right), \tag{157c}$$

$$\chi(z) = (|b| + \Delta)(z - z_{\text{max}}) - \frac{1}{2}\ln\left(\frac{\tilde{\lambda}}{\tilde{\lambda} + (|b| + \Delta)^2}\csc^2\left(\sqrt{\tilde{\lambda}}\,z\right)\right). \tag{157d}$$

The homogeneous solution of the ancillary field is determined using the same procedure as in Appendix C. It is completely fixed for the horizonless solutions once we demand that different parameterizations of the effective action reproducing the same Weyl anomaly should take the same value for a given solution.[15] The coordinate $z$ takes values in the interval $[0, z_{\text{max}}]$ with

$$z_{\text{max}} = \frac{1}{\sqrt{\tilde{\lambda}}}\left(\pi - \cot^{-1}\left(\frac{|b| + \Delta}{\sqrt{\tilde{\lambda}}}\right)\right). \tag{158}$$

Note that the range of this coordinate is affected by backreaction as the dilaton vanishes at a different point. The Ricci scalar is

$$R = -2\left(\tilde{\lambda} + (|b| + \Delta)^2\right), \tag{159}$$

which is the classical result with the shift (152) in $b$.

### 5.3.3 Semiclassical thermodynamics

In the previous subsection, we considered a model where the semiclassical approximation broke down outside of certain temperature regimes. Let us look at whether all classically allowed temperatures are still admissible for the current model. Since the equations can be solved exactly, the only condition follows from the requirement that (147) stays positive for all $z$. Inserting a solution with horizon yields a monotonically decreasing function in $z$ which takes its minimum at the horizon. Thus, we get the condition

$$X_h - 2\Delta\chi_h = |b| + 2\pi T + \Delta\left(1 + 2\ln\frac{T}{T_0}\right) > 0, \tag{160}$$

where $\chi_0 = \ln(8\pi T_0)$. The log term becomes large and negative at small temperatures so we are forced to restrict $T$ to values

$$T > T_0 \exp\left(-\frac{|b|}{2\Delta} - \frac{1}{2}\right). \tag{161}$$

Since $T = 0$ is no longer accessible, the classically allowed solutions with extremal horizons are inconsistent semiclassically. However, when $\Delta \ll |b|$ this lower bound is exponentially small and a near-extremal limit $T \ll |b|/2\pi$ is still possible.

---

[15]There is additional $\Delta$-dependence compared to (231) and (232) because we have not linearized.

Evaluating the action (144) for the solution with horizon (153) gives

$$\Gamma_{\text{eff}} = -\frac{\beta}{2\kappa^2}\left(\frac{2\pi}{\beta} + |b|\right)^2 + \frac{2\pi\Delta}{\kappa^2}\left(1 + 2\chi_h\right), \tag{162a}$$

where $\chi_h$ is the ancillary field evaluated at the horizon. Its value there depends on the temperature-independent integration constant $\chi_0$ and therefore is generically unfixed by our requirements for the semiclassical theory. As in the classical theory, the solution with horizon dominates the canonical ensemble. Its free energy is given by $F = T\Gamma_{\text{eff}}$

$$F = -\frac{1}{2\kappa^2}\left(2\pi T + |b|\right)^2 + \frac{\Delta}{\kappa^2}2\pi T\left(1 + 2\chi_h\right). \tag{163}$$

This leads to an entropy (with $\kappa^2 = 8\pi G$ and $\Delta = NG/3$)

$$S = \frac{X_h}{4G} - \frac{N\chi_h}{6} = \frac{1}{4G_{\text{eff},h}}, \tag{164}$$

where $G_{\text{eff},h}$ is the value of the effective Newton's constant (147) at the horizon. This (macroscopic) result matches with the Wald entropy derived from the effective action (144). The internal energy

$$E = M + \frac{N}{6}T. \tag{165}$$

is expressed in terms of the parameter $M$ appearing in the original Schwarzschild form of the solution (12). As expected, the same result is obtained (holographically) from the Brown–York stress tensor

$$E_{\text{BY}} = \lim_{z\to 0}\frac{2}{\sqrt{h}}\frac{\delta\Gamma_{\text{eff}}}{\delta h_{\tau\tau}}u_\tau u_\tau = M + \frac{N}{6}T, \quad u^\mu = 2e^{-\omega}\delta^\mu_{\ \tau}. \tag{166}$$

Integrating the first law $dE = T\,dS$ yields the (holographic) entropy

$$S_{\text{hol}} = \frac{\pi T}{2G} + \frac{N}{6}\ln T + s_0, \tag{167}$$

with an integration constant $s_0$. Setting

$$s_0 = \frac{|b|}{4G} + \frac{N}{12} - \frac{N}{6}\chi_0 + \frac{N}{6}\ln(8\pi), \tag{168}$$

matches the Wald entropy (164).

For the horizonless solutions, the on-shell action is

$$\Gamma_{\text{eff}} = \frac{\beta}{\kappa^2}\left(\frac{3}{2}\left(\tilde{\lambda} + |b|^2\right) + 2\Delta|b| + \Delta\left(\tilde{\lambda} + (|b| + \Delta)^2\right)z_{\text{max}}\right), \tag{169}$$

with $z_{\text{max}}$ defined in (158). Parameterizing the horizonless solutions in terms of $\tilde{\lambda} > 0$ makes it apparent that the semiclassical correction is strictly positive in models with $b < 0$. Thus, the value of the on-shell action and the associated free energy for the horizonless solutions is positive and larger than in the classical theory.

### 5.3.4  Semiclassical stability

The on-shell action for the solution with horizon is the explicitly negative classical result plus an $\mathcal{O}(\Delta)$ correction that may be positive or negative depending on the sign of $\chi_h$. As in the classical theory, this contribution to the path integral always dominates the contribution from a horizonless solution since (169) is strictly positive. But there is once again a continuum of

horizonless solutions, so we should ask whether their cumulative contributions to the path integral might overwhelm that of the single solution with horizon. Rather than repeating the full analysis of section 3.3, it is sufficient to consider the difference in free energies between a horizonless solution and the solution with horizon. Recall that in the classical theory the smallest value of $\Delta F$ at a given temperature occurs for the horizonless solution $\tilde{\lambda} = 0$, with

$$(\Delta F)_{\text{cl,min}} = 2|b|^2 + 2\pi^2 T^2 + 2\pi|b|T > 2|b|^2. \tag{170}$$

Taking the semiclassical corrections into account, we have (setting $\kappa^2 = 1$)

$$\Delta F = (\Delta F)_{\text{cl,min}} + \frac{3}{2}\tilde{\lambda} + 2\Delta|b| + \Delta\left(\tilde{\lambda} + (|b| + \Delta)^2\right)z_{\text{max}} - 2\pi T \Delta + 4\pi T \Delta \ln\frac{T}{T_0}. \tag{171}$$

As in the analysis of the lower bound on $T$, the constant $\chi_0$ in the ancillary scalar has been written as $\chi_0 = \ln(8\pi T_0)$. There are two differences to the classical result. First, the temperature-dependent terms give a contribution of order $\Delta$ that is negative around a local minimum at $T = T_0/\sqrt{e}$. However, since we assume $\Delta \ll |b|$, this is always much smaller than the positive $2\pi|b|T$ term in (170). Second, the smallest free energy for a horizonless solution occurs at some $\tilde{\lambda} > 0$, rather than $\tilde{\lambda} = 0$. The free energy for the horizonless solution with this value of $\tilde{\lambda}$ is larger than the (positive) classical result by an amount that is of order $\sqrt{\Delta}$. This positive contribution to $\Delta F$ is parametrically larger than the negative contribution from the temperature-dependent correction terms, which was already too small to appreciably change the classical result.

We conclude that semiclassical effects generically enlarge the gap $\Delta F$ between the free energies for the full range of allowed temperatures, so that the stability conclusions of section 3.3 become more robust than in the classical theory.

# 6 Discussion

We summarize briefly the key aspects of our paper. We started with a 2D dilaton gravity model (1) whose specific potential (2) led to solutions of state-dependent constant curvature (15), so that the same model accommodates $dS_2$, Minkowski$_2$, and $AdS_2$ as part of its state space. The state-dependent curvature is associated with the gravitational charges of our theory. This is distinct from other constructions that yield a state-dependent cosmological constant, at the price of introducing extra gauge fields [45–52]. It also differs from the "centaur geometry" studied in [53] (see also [54]), which interpolates between $AdS_2$ in the UV and $dS_2$ in the IR. Unlike the centaur geometries, solutions in our model have constant curvature, whose value and sign are state-dependent.

A curious aspect of our model is the inversion between dilaton field and radial coordinate (13), implying that the weak coupling region $X \to \infty$ corresponds to a center geometrically ($r \to 0$), while the asymptotic region $r \to \infty$ implies strong coupling ($X \to 0$). Apart from these key features, the model is similar to the Almheiri–Polchinski model [10], to which it is conformally related, see (19). The sign of the curvature depends on the mass parameter $\mu$ labeling our solutions and also on the model parameter $b$, see table 1. From the table, it is evident that $dS_2$ [$AdS_2$] appears always at the high-mass [low-mass] end of the spectrum. Thus, $dS_2$ can be viewed as an excitation above $AdS_2$, in essentially the same way that the Schwarzschild black hole is an excitation above Minkowski space.

The Euclidean version of our theory led to interesting thermodynamics, where we defined the canonical ensemble by fixing the temperature at a dilaton isosurface in the weak coupling region. Depending on the sign of one of our model parameters, a unique regular (and perturbatively stable) solution with horizon exists either for any temperature ($b \leq 0$) or for any

temperature above a lower bound ($b > 0$). Additionally, there is a continuum of horizonless states. The free energy of the solution with horizon is negative for any positive temperature (31), while the free energy for the solutions without horizon is non-negative (35). As consequence, the solution with horizon is always the state of lowest free energy, even though it has a higher internal energy (33). Physically, this happens because the huge entropy of these states reduces the free energy more than enough to be competitive with the horizonless states of lower internal energy. The solutions with horizon are locally AdS$_2$ for negative $b$ and at small temperatures, $T < \frac{|b|}{2\pi}$; otherwise they are locally dS$_2$. We also addressed non-perturbative thermodynamical stability in section 3.3 and found that models with negative $b$ are dominated by the solution with horizon for all temperatures. By contrast, for positive $b$ we found a temperature range (51) where the continuum of horizonless states dominates.

To prepare the stage for backreactions, we added scalar matter in section 4. It was convenient to work in conformal gauge and to introduce a fixed auxiliary Poincaré AdS$_2$ spacetime with conformal factor (64), related to the physical metric by a Weyl rescaling (83). This allowed us to solve the equations by standard methods. Associated Penrose diagrams are displayed in Figure 3. As an explicit example, we considered a matter shockwave in section 4.4, yielding the spacetime diagram depicted in Figure 4. It describes the nucleation of a bubble of different spacetime curvature.

In section 5, we analyzed our model semiclassically, taking into account 1-loop effects and backreaction. We worked perturbatively, at small Newton constant and large number of matter fields, while still keeping their product small. An important subtlety was the choice of measure in (96), parametrized by a real constant $\eta$. The standard choice of measure, with $\eta = 0$, was analyzed in detail in section 5.2 where we found that the perturbative regime breaks down at small values of the dilaton. This forced us to exclude the continuum of horizonless solutions in the semiclassical treatment. For $b < 0$ the ground state at a given temperature then turned out to be dominated by a horizon patch of AdS$_2$ at low temperatures and dS$_2$ at high temperatures, just like in the classical model. Although the classical equations of motion match with [10] for the model $b = -1$, the semiclassical theory differs from the one investigated there. However, at low temperatures, we still find a similar behavior of the entropy and internal energy (143). In section 5.3 we investigated the measure $\eta = 2$, which is unique if one wants to treat the continuum of horizonless solutions semiclassically, see Appendix B. As opposed to the standard measure, there are additional ambiguities in the definition of the theory which were fixed by demanding a consistent description in terms of a local effective action (144). One of the crucial ingredients here was to take a certain homogeneous solution for the ancillary field, as explained in Appendix C. We analyzed the thermodynamics of the models with $b < 0$ and argued that semiclassical effects enhance stability of the ground state with horizon, thereby essentially reproducing the classical thermodynamic behavior.

In this paper we took an intrinsically 2D perspective. However, from the viewpoint of dimensional reduction a linear coupling of the matter field to the dilaton is required. Therefore, it could be rewarding to generalize our discussion to non-minimal coupling in (93). Another possible generalization is to relax the assumption of staticity in our treatment of backreaction. This would accommodate time dependent configurations such as the shockwaves in [55].

We conclude with an intriguing observation, and additional prospects for future research. Our analysis shows that the $b < 0$ model has a low-temperature phase dominated by AdS$_2$ and a high-temperature phase dominated by dS$_2$. This result is reminiscent of Susskind's proposal [56] for the Sachdev–Ye–Kitaev (SYK) model [57–59]. While the low temperature phase of the large $N$ limit of SYK has a gravity interpretation in terms of the JT model (see e.g. [11–13, 15, 60–63] and refs. therein), Susskind proposed for its high-temperature phase a gravitational description in terms of dS$_2$. Since our model is conformally related to the AP model, which in turn is related to the JT model by a shift of the dilaton, it is tempting to use

our model as a possible realization of Susskind's proposal.

For applications to SYK-like holographic correspondences, it is necessary to relax the condition that the boundary is a dilaton isosurface. Like for the JT model [64], this requires the addition of new boundary counterterms to the action (3) beyond the usual ones (5). It could be rewarding to apply the covariant phase space analysis of general 2D dilaton gravity [7, 65] to study these and other boundary conditions, and to derive the associated asymptotic (or near horizon) symmetries.

# Acknowledgements

We are grateful to Romain Ruzziconi, Dima Vassilevich and Céline Zwikel for discussions.

**Funding information** This work was supported by the Austrian Science Fund (FWF), projects P 30822, P 32581 and P 33789. Additionally, RM thanks Loyola University Chicago for support via the Faculty Development Leave Program.

# A  Models with state-dependent constant curvature

In this Appendix, we identify models of dilaton gravity in 2D with constant curvature solutions, where the curvature is state-dependent. That is, the curvature is determined by a constant of integration that distinguishes different solutions, rather than fixed parameters appearing in the Lagrangian.

Consider a dilaton gravity model in 2D with bulk Lagrangian

$$L \sim \sqrt{-g} \left[ X R - U(X)(\nabla X)^2 - 2 V(X) \right]. \tag{172}$$

A generalized Birkhoff theorem ensures the existence of a Killing vector, the orbits of which are isocurves of $X$, so that the function $\xi$ in the Schwarzschild-like metric (8) can be expressed as a function of the dilaton. Solutions of the EOM are

$$\xi(X) = e^{Q(X)}(w(X) - 2M), \qquad \partial_r X = \pm e^{-Q(X)}. \tag{173}$$

Here, $M$ is an integration constant, and $Q(X)$ and $w(X)$ are determined by the functions $U(X)$ and $V(X)$ appearing in the bulk Lagrangian

$$Q(X) = Q_0 + \int^X \mathrm{d}y \, U(y), \qquad w(X) = w_0 - 2\int^X \mathrm{d}y \, V(y) e^{Q(y)}. \tag{174}$$

The constants $Q_0$ and $w_0$ can be absorbed into a coordinate rescaling or the parameter $M$, respectively, so we set them to zero without loss of generality. Note that the EOM only determines $\partial_r X$ up to an overall sign in (173). The choice of sign does not affect any of the results in this Appendix, but in the main text we take the minus sign in passing from (11) to (13).

We are interested in models that yield solutions with constant curvature determined by the integration constant $M$ rather than the functions $U(X)$ and $V(X)$. For the metric (8), the scalar curvature is

$$R = -\partial_r{}^2 \xi(X). \tag{175}$$

Using the solution (173) this can be written as

$$R = 2 M Q'' e^{-Q} - e^{-Q} \left( Q'' w + Q' w' + w'' \right), \tag{176}$$

where a prime indicates a derivative with respect to $X$. Thus, we must find functions $U(X)$ and $V(X)$ such that $Q(X)$ and $w(X)$, as defined in (174), satisfy

$$Q'' e^{-Q} = \lambda, \qquad Q'' w + Q' w' + w'' = 0, \tag{177}$$

for some constant $\lambda$.

The first equation in (177) is solved by expressing $U$ as a function of $Q(X)$, which yields

$$\frac{1}{2} \frac{\partial}{\partial Q} (U^2) = \lambda e^Q. \tag{178}$$

Integrating and using the result to solve $Q'(X) = U(X)$ we arrive at a three-parameter set of solutions for $U(X)$. After investigating various solutions, we find that the essential features are captured by the simplest member of this family

$$U(X) = -\frac{2}{X}. \tag{179}$$

Then,

$$Q = -2 \ln X \tag{180}$$

and $e^Q = X^{-2}$. For this solution $\lambda = 2$. The second equation in (177) yields the Weyl invariant function $w(X) = b_1 X^2 + 2 b_2 X$, where $b_1$ and $b_2$ are arbitrary constants, which is the same result as for the AP model. The definition (174) of $w(X)$ gives the function $V(X)$ appearing in the Lagrangian

$$V(X) = -b_1 X^3 - b_2 X^2. \tag{181}$$

These results for $U(X)$ and $V(X)$ are our starting point in section 2, the action (3). The resulting model has solutions with state-dependent curvature, $R = 4M$.

## B  Linearized solution of semiclassical equations of motion

In this Appendix, we solve the EOM for the class of semiclassical theories given by the gravity action in (29), with matter coupling (93) and the path integral measure (96) together with $\eta < 3$. As discussed in section 5.1 the Weyl anomaly is given by the components

$$\langle T_{+-} \rangle = -\frac{N}{24\pi} \partial_+ \partial_- \left(2\omega + \eta \ln X\right), \tag{182a}$$

$$\langle T_{\pm\pm} \rangle = \frac{N}{24\pi} \Big(2\partial_\pm^2 \omega - 2(\partial_\pm \omega)^2 + \eta \, \partial_\pm^2 \ln X,$$

$$- \gamma \, (\partial_\pm \ln X)^2 - 2\eta \, \partial_\pm \omega \, \partial_\pm \ln X + (\partial_\pm k(X))^2\Big) + \tau_{\pm\pm}(x^\pm), \tag{182b}$$

$$\langle T_X \rangle = \frac{N}{3\pi} e^{-2\omega} \frac{1}{X} \partial_+ \partial_- \left(\eta \, \omega + \gamma \ln X\right) - \frac{N}{3\pi} e^{-2\omega} k'(X) \partial_+ \partial_- k(X), \tag{182c}$$

where $k(X)$ and $\gamma$ represent ambiguities in the definition of these components. It is convenient to express the semiclassical EOM in terms of the variable $\hat\omega = \omega + \ln X$ described in section 2.3. After some simplifications this leads to

$$0 = \partial_+ \partial_- \hat\omega + \frac{1}{4} e^{2\hat\omega} + \kappa^2 \Big(\frac{1}{X} \langle T_{+-} \rangle + \frac{1}{8X^2} e^{2\hat\omega} \langle T_X \rangle\Big), \tag{183a}$$

$$0 = -\partial_+ \partial_- X - \frac{1}{2} e^{2\hat\omega} (X + b) + \kappa^2 \langle T_{+-} \rangle, \tag{183b}$$

$$0 = e^{2\hat\omega} \partial_\pm \left(e^{-2\hat\omega} \partial_\pm X\right) + \kappa^2 \langle T_{\pm\pm} \rangle. \tag{183c}$$

For specificity, we restrict to models with $b = 0$, but the procedure is analogous for $b \neq 0$. We make the ansatz $\hat{\omega} = \hat{\omega}_B + \Delta \, \delta \hat{\omega}$, $X = X_B + \Delta \, \delta X$ with $\hat{\omega}_B$ and $X_B$ satisfying the above equations with the sources $\langle T_{\mu\nu} \rangle$ and $\langle T_X \rangle$ set to zero. Expanding (183) to linear order in $\Delta$ yields

$$0 = \partial_+ \partial_- \delta \hat{\omega} + \frac{1}{2} e^{2\hat{\omega}_B} \delta \hat{\omega} + \left( \frac{1}{X} \Theta_{+-} + \frac{1}{8X^2} e^{2\hat{\omega}} \Theta_X \right), \tag{184a}$$

$$0 = \partial_+ \partial_- \delta X + \frac{1}{2} e^{2\hat{\omega}_B} \delta X + e^{2\hat{\omega}_B} X_B \delta \hat{\omega} - \Theta_{+-}, \tag{184b}$$

$$0 = \partial_\pm^2 \delta X - 2\partial_\pm \delta \hat{\omega} \partial_\pm X_B - 2\partial_\pm \hat{\omega}_B \partial_\pm \delta X + \Theta_{\pm\pm}. \tag{184c}$$

The rescaled components of the anomalies $\Theta_{\mu\nu} := \langle T_{\mu\nu} \rangle \frac{\kappa^2}{\Delta}$ and $\Theta_X := \langle T_X \rangle \frac{\kappa^2}{\Delta}$ are evaluated using the background fields $\hat{\omega}_B$ and $X_B$, since we are linearizing. As we are only interested in static configurations we switch to coordinates

$$ds^2 = \frac{1}{4} e^{2\omega} \left( d\tau^2 + dz^2 \right), \tag{185}$$

and simplify the EOM by setting $\partial_\pm = \pm \partial_z$. The linearized EOM (184a) and (184b) take the same general form

$$\partial_z^2 y(z) - \frac{1}{2} e^{2\hat{\omega}_B} y(z) = j_y(z), \tag{186}$$

where $y(z)$ is either $\delta \hat{\omega}$ or $\delta X$. The source term $j_{\delta \hat{\omega}}$ depends on the background fields, while $j_{\delta X}$ depends on both the background fields and $\delta \hat{\omega}$.

The linearized EOM can be solved analytically to determine the semiclassical corrections $\delta \hat{\omega}(z)$ and $\delta X(z)$. The correction to the physical metric is then obtained by transforming back to unhatted variables

$$\delta \omega = \delta \hat{\omega} - \frac{\delta X}{X_B}. \tag{187}$$

The constraints (184c) will subsequently determine the integration functions $\tau_{\pm\pm} = \tau$ which are constant for a static configuration. Solving the equations (184a) and (184b) introduces four integration constants $\{c_{\hat{\omega}1}, c_{\hat{\omega}2}, c_{X1}, c_{X2}\}$ as coefficients of the homogeneous solutions $y_1(z)$ and $y_2(z)$ of (186):

$$\delta \hat{\omega}_{\text{hom}} = c_{\hat{\omega}1} \, y_1(z) + c_{\hat{\omega}2} \, y_2(z), \tag{188}$$

$$\delta X_{\text{hom}} = c_{X1} \, y_1(z) + c_{X2} \, y_2(z). \tag{189}$$

Just like in Schwarzschild gauge the spatial coordinate $z$ has a range $z \in [z_{\text{min}}, z_{\text{max}}]$, with the values depending on the solution we are considering. When linearizing around a background solution with horizon, two of the four constants in the homogeneous solutions are fixed by the conditions that scalar fluctuations are small,

$$\frac{\delta X}{X_B} \overset{!}{\ll} 1 \quad \text{and} \quad \frac{\delta R}{R_B} \overset{!}{\ll} 1, \qquad \forall z \in [z_{\text{min}}, z_{\text{max}}]. \tag{190}$$

A third constant can be absorbed into a constant rescaling of the coordinates in (185), and the fourth is fixed by regularity at the horizon. However, when attempting to linearize around backgrounds without horizon we encounter an obstruction that cannot be removed by an appropriate choice of constants in the homogeneous solutions. In section B.2 we will show that only a single value of $\eta$ in the allowed range avoids this obstruction. That case corresponds to a model where the EOM and constraints become (with some assumptions) exactly solvable, see section 5.3.

### B.1 Background with horizon

As discussed in section 2.2 the $b = 0$ model has vacuum solutions with horizon for any boundary conditions $\beta > 0$. We can transform (21) to the gauge (185) by

$$z = \frac{1}{\sqrt{\mu}} \coth^{-1}\left(\frac{1}{\sqrt{\mu}\, r}\right), \tag{191}$$

mapping to the range $z \in [0, \infty)$ where $z \to \infty$ corresponds to the horizon and $z \to 0$ to the weak coupling region (the $X \to \infty$ boundary). The background solutions take the form

$$X_B(z) = \sqrt{\mu} \coth(\sqrt{\mu}z), \tag{192a}$$

$$\omega_B(z) = \frac{1}{2}\ln\left(\frac{4}{\cosh^2(\sqrt{\mu}z)}\right), \tag{192b}$$

$$\hat{\omega}_B(z) = \omega_B + \ln X_B = \frac{1}{2}\ln\left(\frac{4\mu}{\sinh^2(\sqrt{\mu}z)}\right), \tag{192c}$$

and the homogeneous solutions of (186) are

$$y_1(z) = \coth(\sqrt{\mu}z), \qquad y_2(z) = 1 - \sqrt{\mu}z \coth(\sqrt{\mu}z). \tag{193}$$

Let us set the arbitrary function $k(X)$ in (182) to zero for a moment and see what the results are in that case. Plugging the background into (184) we solve as described above obtaining

$$\delta\omega(z) = c_{\hat{\omega}2} - \frac{c_{X1}}{\sqrt{\mu}} + \frac{\tanh(\sqrt{\mu}z)}{2\sqrt{\mu}}\left(3\gamma - 4\eta - 2c_{\hat{\omega}2}\mu z + 2(\eta-2)\ln(\cosh(\sqrt{\mu}z))\right), \tag{194}$$

$$\delta X(z) = 1 - \gamma + \eta - c_{X1}\coth(\sqrt{\mu}z)$$
$$\qquad - c_{\hat{\omega}2}\mu z\, \text{csch}^2(\sqrt{\mu}z) + (\eta-2)\text{csch}^2(\sqrt{\mu}z)\ln(\cosh(\sqrt{\mu}z)), \tag{195}$$

$$\delta R(z) = \sqrt{\mu}\left(4c_{X1} - 6(\eta-2)\tanh(\sqrt{\mu}z)\right). \tag{196}$$

The consistent linearization condition (190) has been used to fix two of the four integration constants in (188) to the values

$$c_{X2} = 0, \qquad c_{\hat{\omega}1} = \frac{\gamma-2}{2\sqrt{\mu}}. \tag{197}$$

At $z = 0$ the line element for the background solution takes the form $ds^2 \simeq d\tau^2 + dz^2$. This receives a constant rescaling at order $\Delta$ from the linearized solution, which can be canceled by fixing

$$c_{\hat{\omega}2} = \frac{1}{\sqrt{\mu}}c_{X1}. \tag{198}$$

Finally, the regularity condition at the horizon gives

$$\beta = \lim_{z\to\infty}\left(-\frac{2\pi}{\partial_z\omega}\right) = \frac{2\pi}{\sqrt{\mu}}\left(1 + \Delta\left(\frac{\eta-2}{\sqrt{\mu}} - c_{X1}\right)\right). \tag{199}$$

This can be solved by setting $\mu = 2\pi/\beta$, as with the background solution, and

$$c_{X1} = \eta - 2. \tag{200}$$

Equivalently, since $\mu$ is just an integration constant, we can absorb the $c_{X1}$ term via the rescaling[16] $\mu \to \mu(1 - 2\Delta c_{X1}/\sqrt{\mu})$. Then to linear order in $\Delta$ the condition (199) fixes $\mu$ to the value

$$\mu = \left(\frac{2\pi}{\beta}\right)^2 \left(1 + \Delta \frac{\beta(\eta - 2)}{\pi}\right). \tag{201}$$

Either approach gives the same result. It follows that for any allowed values of $\eta$ and $\gamma$ we can always find a consistent and unique linearized solution around the background with horizon. Let us now turn to solutions without horizon.

## B.2  Background without horizon

Solutions without horizon have $\mu < 0$ and describe locally AdS$_2$ spacetimes [cf. (17)]. Defining the positive quantity $\lambda := |\mu|$ and analytically continuing (191), we find the background solution

$$X_B(z) = \sqrt{\lambda}\cot(\sqrt{\lambda}z), \tag{202a}$$

$$\omega_B(z) = \frac{1}{2}\ln\frac{4}{\cos^2(\sqrt{\lambda}z)}, \tag{202b}$$

$$\hat{\omega}_B(z) = \omega_B + \ln X_B = \frac{1}{2}\ln\frac{4\lambda}{\sin^2(\sqrt{\lambda}z)}. \tag{202c}$$

The radial coordinate is chosen to take values $z \in [0, \pi/(2\sqrt{\lambda}))$ where again $z \to 0$ corresponds to the weak-coupling region ($X_B \to \infty$) and $z \to \pi/(2\sqrt{\lambda})$ is the region of vanishing background dilaton. We proceed as before with solving (184), setting $k(X) = 0$ for now. The expressions we get for $\delta X$ and $\delta R$ are

$$\delta X = 1 - \gamma + \eta - c_{X1}\bar{y}_1(z) + c_{X2}\bar{y}_2(z) + \csc^2(\sqrt{\lambda}z)\sqrt{\lambda}\left(c_{\hat{\omega}1} - c_{\hat{\omega}2}\sqrt{\lambda}z\right)$$

$$+ \frac{\csc^2(\sqrt{\lambda}z)}{2}\left(\gamma - 2 + 2(2 - \eta)\ln(\cos(\sqrt{\lambda}z))\right), \tag{203}$$

$$\delta R = 6(\eta - 2)\sqrt{\lambda}\tan(\sqrt{\lambda}z) - 4c_{X1}\sqrt{\lambda} + 4c_{X2}z\lambda. \tag{204}$$

Here, $\bar{y}_i(z)$ are the analytically continued versions of homogeneous solutions (193)

$$\bar{y}_1(z) = \cot(\sqrt{\lambda}z), \quad \bar{y}_2(z) = 1 - \sqrt{\lambda}z\cot(\sqrt{\lambda}z). \tag{205}$$

To satisfy the first linearization condition (190) $\delta X$ must approach zero at least as rapidly as $X_B$ for $z \to z_{\max} = \frac{\pi}{2\sqrt{\lambda}}$. Expanding around that value

$$\lim_{z \to z_{\max}} \delta X = (2 - \eta)\ln\left(\sqrt{\lambda}(z_{\max} - z)\right) + c_{X2} - \frac{\gamma}{2} + \eta + \frac{\sqrt{\lambda}}{2}(2c_{\hat{\omega}1} - c_{\hat{\omega}2}\pi) + \mathcal{O}(z_{\max} - z), \tag{206}$$

it becomes evident that a logarithmic divergence appears at small values of the background dilaton which cannot be cancelled by any choice of the homogeneous solution. We thus have to fix $\eta = 2$ to make sense of linearized backreaction for horizonless states as only then the corrections have the chance to be small compared to the background. A similar problem would appear for the correction of the Ricci scalar (204). However, the divergent term automatically cancels for the above choice of $\eta$.

---

[16]The background dilaton is $X_B(z) = \sqrt{\mu}\,y_1(z)$, so the coefficient $c_{X1}$ of $y_1(z)$ in the homogeneous solution $\delta X_{\text{hom}}$ can naturally be absorbed in this manner.

One may ask, whether instead of fixing $\eta = 2$ one can instead make an appropriate choice of $k(X)$. We argue now that this is not possible. In the following three examples the same system of equations is solved for specific choices of $k(X)$ including an arbitrary coefficient $\rho \in \mathbb{R}$. The corrections to $X$ and $R$ then take the form

$$k(X) = \rho X: \qquad \lim_{z \to z_{\max}} \delta X = (2 - \eta) \ln\left(\sqrt{\lambda}(z_{\max} - z)\right) + \frac{\lambda \rho^2}{6} + \mathcal{O}(1), \tag{207}$$

$$\delta R = \delta R\big|_{\rho=0} + \frac{2\lambda^{\frac{3}{2}}\rho^2}{3} \cot(\sqrt{\lambda}z)\left(8 - 5\csc^2(\sqrt{\lambda}z)\right), \tag{208}$$

$$k(X) = \rho \ln X: \qquad \lim_{z \to z_{\max}} \delta X = (2 - \eta) \ln\left(\sqrt{\lambda}(z_{\max} - z)\right) + \frac{\rho^2}{2} + \mathcal{O}(1), \tag{209}$$

$$\delta R = \delta R\big|_{\rho=0}, \tag{210}$$

$$k(X) = \frac{\rho}{X}: \qquad \lim_{z \to z_{\max}} \delta X = \left(2 - \eta + \frac{2\rho^2}{3\lambda}\right) \ln\left(\sqrt{\lambda}(z_{\max} - z)\right) + \mathcal{O}(1), \tag{211}$$

$$\delta R = \delta R\big|_{\rho=0} - \frac{2\rho^2}{3\sqrt{\lambda}} \tan(\sqrt{\lambda}z)\left(8 - 5\sec^2(\sqrt{\lambda}z)\right). \tag{212}$$

In each case the behavior of $\delta X$ is shown in the problematic limit $z \to z_{\max}$.

In (207) and (209) the log divergence in $\delta X$ is unaffected. A similar result is obtained if $k(X)$ is any positive power of $X$. So for the first two choices of $k(X)$ we cannot get around setting $\eta = 2$. In addition, $\delta R$ in (208) has a divergence at $z \to 0$, which violates the second linearization condition in (190). The choice $k(X) = \rho/X$ changes the logarithmic term in (211), but to cancel the divergence we would have to choose $\rho = \rho(\lambda)$. This clearly does not make sense as $\rho$ is a parameter of the model and thus cannot be dependent on the solution. And, as with positive powers of $X$, this choice introduces a divergence in $\delta R$, this time at $z \to z_{\max}$. Other negative powers of $X$ lead to the same behavior, so choices of $k \sim X^c$ with $c < 0$ have to be excluded as well. The only consistent choice is $k(X) = \rho \ln X$. But looking at (182b) it becomes clear that this just amounts to a shift $\gamma \mapsto \gamma - \rho^2$. Therefore, $k(X)$ can always be absorbed by a redefinition of $\gamma$.

We conclude that a consistent linearized backreaction for the horizonless solution with general values of $\eta$ is not possible. Under these assumptions, the choice $\eta = 2$ is the unique measure allowing a consistent backreaction for solutions without horizon.

## C  One-loop effective action

In this Appendix, we construct a local description of the effective action for the semiclassical models studied in sections 5.2 and 5.3. For the $\eta = 0$ model this is straightforward, but there are subtleties when writing down a local effective field theory description that reproduces the semiclassical theory with $\eta = 2$ and $k(X) = 0$. Demanding the existence of such a description fixes the last remaining free parameter $\gamma$.

The effective action obtained by integrating out the matter fields in the path integral is in general non-local. Along the lines of [10, 20] it should, however, be possible to obtain an equivalent action which is local, diffeomorphism invariant, and second-order in derivatives of the fields by adding a new "ancillary" scalar field $\chi$. Evaluating this action on a solution of the EOM for $\chi$ gives the original non-local action, and the coupling of $\chi$ to the metric and dilaton reproduces the semiclassical corrections $\langle T_{\mu\nu} \rangle$ and $\langle T_X \rangle$ to the EOM. The action has the form

$$\Gamma_{\text{eff}}[g, X, \chi] = \Gamma_B + \Gamma_\chi, \tag{213}$$

where $\Gamma_{\!B}$ is the classical Euclidean action (29) with $f_i = 0$, and $\Gamma_{\chi}$ is the action for the ancillary field that encodes semiclassical effects.

For the $\eta = 0$ model considered in section 5.2, an action $\Gamma_{\chi}$ that satisfies our requirements is

$$\Gamma_{\chi} = \frac{\Delta}{\kappa^2} \int_M d^2x \sqrt{g}\left(\chi R + (\nabla \chi)^2\right) + \frac{2\Delta}{\kappa^2} \int_{\Sigma} dx \sqrt{h}\, \chi\, K\,. \tag{214}$$

This is proportional to the parameter $\Delta = N G / 3$ defined in (94). The EOM obtained by varying the action with respect to $\chi$ is

$$2\nabla^2 \chi = R\,. \tag{215}$$

The variation of $\Gamma_{\chi}$ with respect to the metric, evaluated on a solution of the $\chi$ EOM, reproduces the semiclassical corrections studied in section 5.2.

In the rest of this appendix we focus exclusively on the model with $\eta = 2$ and $k(X) = 0$ studied in section 5.3. This requires a more general action that includes different possible couplings between $\chi$ and the dilaton. The two-derivative terms that might arise are

$$\Gamma_{\chi} = \frac{\Delta}{\kappa^2} \int_M d^2x \sqrt{g}\left(\chi R + (\nabla \chi)^2 + c_4 R \ln X + c_5 (\nabla \ln X)^2\right. \tag{216}$$

$$\left. + c_1 \chi \nabla^2 \ln X + c_2 \nabla \chi \nabla \ln X + c_3 \ln X \nabla^2 \chi\right)$$

$$+ \frac{2\Delta}{\kappa^2} \int_{\Sigma} dx \sqrt{h}\left((\chi + c_4 \ln X) K - \frac{c_1}{2} \chi n^{\mu} \nabla_{\mu} \ln X - \frac{c_3}{2} \ln X\, n^{\mu} \nabla_{\mu} \chi + \mathcal{L}_{\mathrm{ct}}\right).$$

This action includes a boundary integral with Gibbons–Hawking–York contributions and other terms required by second-derivative bulk terms. There is also a boundary counterterm Lagrangian $\mathcal{L}_{\mathrm{ct}}$ that is fixed once we have determined the boundary conditions for the ancillary field. The coefficients $c_i$ in the bulk part of $\Gamma_{\chi}$ parameterize the possible effective terms consistent with the above requirements.

The EOM for $\chi$ obtained by varying (216) is

$$2\nabla^2 \chi = R + \left(c_1 - c_2 + c_3\right) \nabla^2 \ln X\,. \tag{217}$$

Solving this equation in conformal gauge (9) gives

$$\chi = -\omega + \frac{c_1 - c_2 + c_3}{2} \ln X + \chi^+(x^+) + \chi^-(x^-)\,, \tag{218}$$

with two unspecified integration functions $\chi^{\pm}(x^{\pm})$. Inserting this result into

$$\langle T^{\mu\nu}\rangle = \frac{2}{\sqrt{g}} \frac{\delta \Gamma_{\chi}}{\delta g_{\mu\nu}}\,, \quad \langle T_X\rangle = \frac{2}{\sqrt{g}} \frac{\delta \Gamma_{\chi}}{\delta X}\,, \tag{219}$$

obtains (105), (109) and (110) with $k(X) = 0$ and the identification

$$\eta = -c_1 + c_2 - c_3 - 2c_4\,, \quad \gamma = \frac{(c_1 - c_2 + c_3)^2}{2} - 2c_5\,. \tag{220}$$

The $\tau_{\pm\pm}$ terms in $\langle T_{\pm\pm}\rangle$ are related to the functions $\chi^{\pm}$ by

$$\tau_{\pm\pm}(x^{\pm}) = \frac{N}{24\pi}\left(2\partial_{\pm}^2 \chi^{\pm} - 2(\partial_{\pm}\chi^{\pm})^2\right). \tag{221}$$

Fixing $\eta = 2$ gives one condition,

$$-c_1 + c_2 - c_3 - 2c_4 = 2\,, \tag{222}$$

on the coefficients in the effective action. But there is still considerable freedom in the choice of action functional. Even if we fix $\gamma$ there is a three-parameter family of actions and one would assume that these might take different values on-shell.

Our goal is to study the semiclassical thermodynamics by evaluating the action on the backreacted solutions, so any ambiguity in the action introduced by this procedure is unsatisfactory. Of course, the point of introducing $\chi$ was just to obtain a convenient local action. We imagine that (up to standard ambiguities in the effective action) there is a unique action we are trying to reproduce. It is therefore natural to demand that the on-shell value of both the action and its first variation be independent of the choice of the parameters $c_i$ once $\eta$ and $\gamma$ are fixed. That such a possibility exists is by no means obvious. However, it turns out to be true in our case.

In section 5.2 the semiclassical EOM could be solved exactly for the model with $\eta = 0$. With the choice of measure $\eta = 2$, this is no longer possible for general values of the parameter $\gamma$. It is therefore necessary to linearize in $\Delta \ll 1$ and repeat the computation of Appendix B for $b \neq 0$. We then evaluate the action (213) on the linearized solution and ask what conditions should hold in order for the result to be independent of the choice of $c_i$ once $\eta$ is fixed. Note that, since we are linearizing, the $\Gamma_\chi$ part of the action is evaluated only on the background solutions. We first consider linearizing around the horizonless AdS solutions described in section 2.2 and (27), and focus on models with $b < 0$ for simplicity. Working in conformal gauge, the horizonless background solutions have $\mu < 0$ and take the form[17]

$$X_B(z) = |b| + \sqrt{|\mu|}\cot\left(\sqrt{|\mu|}\,z\right), \tag{223}$$

$$\omega_B(z) = \frac{1}{2}\ln\frac{4|\mu|}{\left(\sqrt{|\mu|}\cos(\sqrt{|\mu|}\,z) + |b|\sin(\sqrt{|\mu|}\,z)\right)^2}\,, \tag{224}$$

where $z \in [0, z_{\max}]$. The dilaton vanishes at

$$z_{\max} = \frac{1}{\sqrt{|\mu|}}\left(\pi - \cot^{-1}\frac{|b|}{\sqrt{|\mu|}}\right), \tag{225}$$

while $z \to 0$ is the boundary $X \to \infty$. The ancillary field takes the form

$$\chi = -\omega_B + \frac{c_1 - c_2 + c_3}{2}\ln X + \sqrt{|\mu|}\,\chi_1 z + \chi_0\,, \tag{226}$$

where staticity restricts the homogeneous solution to a linear function in $z$ dependent on two integration constants $\chi_0, \chi_1 \in \mathbb{R}$.

Let us consider the various terms in (216). A short calculation shows that contributions at $z \to z_{\max}$ from the $R\ln X$ and $(\partial \ln X)^2$ terms diverge, so we henceforth set $c_4 = 0$ and $c_5 = 0$. In that case the condition (222) on the remaining coefficients becomes $-c_1 + c_2 - c_3 = 2$, and the terms in the second line of (216) can be rewritten as

$$\int_M \mathrm{d}^2 x \sqrt{g}\left(c_1 \chi \nabla^2 \ln X + c_2 \nabla\chi\nabla\ln X + c_3 \ln X \nabla^2 \chi\right) =$$
$$\int_M \mathrm{d}^2 x \sqrt{g}\left(2\,\nabla\chi\nabla\ln X\right) + \left(c_1 B_1 + c_3 B_3\right)\Big|_{z=0}^{z_{\max}}, \tag{227}$$

---

[17]Models with $b \geq 0$ can be treated in the same way. However, for $b > 0$ one must consider solutions involving trig functions of $\sqrt{|\mu|}z$ when $\mu < 0$ and hyperbolic trig functions of $\sqrt{\mu}z$ when $0 \leq \mu < b^2$.

where $B_1$ and $B_3$ are total derivative terms arising from integration-by-parts. Boundary terms in the last line of (216) cancel the contributions from these total derivative terms at $z \to 0$, leaving just the contributions at $z \to z_{max}$

$$B_1\Big|_{z_{max}} \propto \lim_{z \to z_{max}} \beta \sqrt{h}\, \chi\, n^\mu \partial_\mu \ln X \,, \tag{228}$$

$$B_3\Big|_{z_{max}} \propto \lim_{z \to z_{max}} \beta \sqrt{h}\, \ln X\, n^\mu \partial_\mu \chi \,. \tag{229}$$

These are both non-zero for generic values of $\chi_0$ and $\chi_1$ in (226). Since they are multiplied by $c_1$ and $c_3$ in (227), the on-shell value of the action (216) will depend on the specific choice of coefficients $c_i$ unless the homogeneous solution for $\chi$ is chosen so that both terms vanish. This is accomplished by taking the unique homogeneous solution for $\chi$ such that $\chi$ and $\partial_z \chi$ both go to zero at $z \to z_{max}$. We have the requirements

$$-c_1 + c_2 - c_3 = 2 \,, \tag{230}$$

$$\chi_0 = -|b|\, z_{max} + \frac{1}{2} \ln\left(4(|\mu| + b^2)\right), \tag{231}$$

$$\chi_1 = \frac{|b|}{\sqrt{|\mu|}} \,, \tag{232}$$

that ensure that any particular choice of the parameters $c_1$, $c_2$ and $c_3$ in $\Gamma_\chi$ gives the same on-shell value for the action and its first variation. Finally, an analysis of the variational properties of the action fixes the boundary counterterm in (216) to be

$$\mathcal{L}_{ct} = \frac{X}{2} \,. \tag{233}$$

This boundary term also guarantees that the action is finite on solutions of the semiclassical EOM. A convenient representative of this family of equivalent actions is

$$\Gamma_\chi = \frac{\Delta}{\kappa^2} \int_M d^2x \sqrt{g} \left(\chi R + (\nabla \chi)^2 + 2\nabla \chi \nabla \ln X\right) + \frac{2\Delta}{\kappa^2} \int_\Sigma dx \sqrt{h} \left(\chi K + \frac{X}{2}\right). \tag{234}$$

This form of the action is used throughout section 5.3.

We evaluate this action for solutions with horizon to check that its on-shell value remains independent of the choice of constants $c_i$. The background reads

$$X_B(z) = |b| + \sqrt{\mu} \coth\left(\sqrt{\mu}\, z\right), \tag{235}$$

$$\omega_B(z) = \frac{1}{2} \ln \frac{4\mu}{\left(\sqrt{\mu} \cosh(\sqrt{\mu}\, z) + |b| \sinh(\sqrt{\mu}\, z)\right)^2} \,, \tag{236}$$

where $z \in [0, \infty)$, $\mu \geq b^2$, and the horizon is at $z \to \infty$. With $-c_1 + c_2 - c_3 = 2$ the ancillary field is

$$\chi = -\omega_B - \ln X + \sqrt{\mu}\, \chi_1 z + \chi_0 \,, \tag{237}$$

where the two constants $\chi_0$ and $\chi_1$ in the homogeneous solution do not necessarily take the same values as for the horizonless background. Indeed, evaluating the total derivative contributions (228) and (229) for this solution (at $z \to \infty$) both terms vanish iff $\chi_1 = -1$. There is no condition on $\chi_0$, which can be rewritten in terms of the value $\chi_h$ that the ancillary field takes at the horizon

$$\chi = \chi_h - \sqrt{\mu}\, z + \ln\left(2 \sinh(\sqrt{\mu}\, z)\right). \tag{238}$$

Thus, the on-shell action for solutions with horizon gives the same value for any choice of the constants $c_1$, $c_2$, and $c_3$, subject to the conditions found previously. The local form of the effective action (234) gives the unique (up to standard ambiguities associated with effective actions and boundary counterterms) on-shell value of the action and its first variation for all the solutions we consider.

The action derived above gives a well-defined variational principle in the sense mentioned in section 2.1 and studied in [31]. That is, the first variation of the action vanishes on-shell for all field variations with the same $X \to \infty$ asymptotic behavior as solutions of the EOM. In the coordinates used here this corresponds to $z \to 0$. This means that the corrections to $X$ and $g_{\mu\nu}$ should satisfy the same boundary conditions at $\Sigma$ as the background. To determine the appropriate boundary condition for the ancillary field we expand near $z \to 0$ and find

$$\lim_{z \to 0} \chi = \ln \frac{z}{2} + \chi_0 + \chi_1 \sqrt{\mu} z + \mathcal{O}(z^2), \tag{239}$$

for solutions with horizon, and a similar expression (with $\mu$ replaced by the appropriate parameter) for horizonless solutions. As we have to fix $\chi_1$ and possibly $\chi_0$ to get a consistent on-shell action in the sense mentioned above, we choose the boundary condition of the ancillary field as

$$\delta \chi = \mathcal{O}(z^2). \tag{240}$$

It follows that

$$\delta \Gamma_{\text{eff}} \Big|_{\text{EOM}} = 0, \tag{241}$$

as required. Furthermore, varying the action with respect to the boundary value of the metric gives a finite quasilocal stress tensor.

It is interesting to note that, once we set $c_4$ and $c_5$ to zero, the constraint (220) on the coefficients $c_i$ implies

$$\gamma = 2. \tag{242}$$

In fact, this value for $\gamma$ makes it possible to solve the semiclassical equations exactly, allowing us in section 5.3 to go beyond the linearized solutions considered in this Appendix.

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
