# Peer review of "dS$\boldsymbol{_2}$ as excitation of AdS$\boldsymbol{_2}$"

_SciPost Physics, doi:SciPost Phys. 13, 119 (2022)_

## Round 1 · Referee Report · Anonymous (Referee 1) · 2022-6-29

Report

Referee report on 2204.00045v1

The authors present a general family of classically solvable 2d gravity models. They relate this family to existing models in a detailed manner and emphasize its distinguishing features. Notably, the models do not have any known higher dimensional pedigree. These models allow for a transition between de Sitter and anti-de Sitter, depending on values of the temperature. The dominant state at low temperatures is found to be anti-de Sitter and de Sitter is found at high temperatures.

Remarkably, these features persist to the semi-classical regime, as checked by a linearized analysis and also stability is checked.

Regarding the semi-classical corrections, it is noteworthy that the authors invest effort to go beyond the usual purely conformal matter contribution and also consider taking into account corrections to the dilaton.

The manuscript is written in a clear manner, although fairly technical. My impression is that the results are not too controversial, but thoroughly executed. The manuscript opens pathways to new research, notably the state dependent curvature and semi-classical corrections in 2d models. Before recommending this manuscript for publication, I have some additional questions/comments:

-The authors point out the strongly coupled region at the conformal boundary, with which I am concerned. Indeed, traditionally one would expect a singularity in such a region. Towards the end of page 8 you mention that adding a topological term would make $X=0$ fine. However, $X$ will still give you a diverging value for your effective Newton's constant at some negative value, which is not a ruled out value. How can you defend your semi-classical analysis in that region? It is tempting to make the identification you make relating to SYK, but how do you reconcile that with strong gravity?

-How does your work compare to the ``Centaur'' model in \href{https://arxiv.org/abs/1703.04622}{1703.04622} and its conclusions? There the authors also find a de Sitter realization embedded into anti-de Sitter whilst also having a strongly coupled region near the conformal boundary.

  • Above (118) you make an initial choice fixing $\tau_{\pm\pm}$, which corresponds to some heat bath. This, I believe, is the reason that you not find evaporation of any kind after applying shockwaves in the semi-classical case, see e.g. \href{https://arxiv.org/abs/2004.14944}{2004.14944}. Could you comment on this?

  • Connected to the previous point, making a different choice for $\tau_{\pm\pm}$, it seems reasonable to get evaporation after performing a shockwave. Can you comment on this situation?

  • Is there any ``valid'' value $\eta$ that corresponds to a minimally coupled free scalar field in, e.g., 4d?

  • There have recently been some effort to use Quantum Extremal Surfaces or island formula'' to investigate unitarity of semi-classical de Sitter models using the Page curve. Could you study islands in your model, or do you expect that the strongly coupled region stops you from, e.g.,attaching'' a bath region needed for collection of radiation?

  • validity: -
  • significance: -
  • originality: -
  • clarity: -
  • formatting: -
  • grammar: -

Author:  Robert McNees  on 2022-07-14  [id 2660]

(in reply to Report 1 on 2022-06-29)

We would like to thank the referee for their insightful questions and useful feedback. Below, we address their questions and comments point-by-point. At the end of our responses is a list of changes made in a new version of the paper that is now available on the arXiv.

The referee writes: The authors point out the strongly coupled region at the conformal boundary, with which I am concerned. Indeed, traditionally one would expect a singularity in such a region. Towards the end of page 8 you mention that adding a topological term would make $X = 0$ fine. However, X will still give you a diverging value for your effective Newton's constant at some negative value, which is not a ruled out value. How can you defend your semi-classical analysis in that region? It is tempting to make the identification you make relating to SYK, but how do you reconcile that with strong gravity?

Our response: Our analysis is restricted to the region $X \geq 0$. From a 2D perspective this restriction is natural, because $X = 0$ is the conformal boundary of the spacetime. Continuing to negative values of X would correspond to moving "beyond" the conformal boundary, which is infinitely far away from points in the bulk ($X > 0$) of the spacetime.

As far as making the identification with SYK, this is an excellent question.

The comment about SYK in the Discussion is based on two observations. First, our model is conformally related to models that are dual to SYK and SYK-like theories. Second, in our model with $b < 0$, there is a transition on the gravity side from AdS$_2$ at low temperature to dS$_2$ at high temperature. This is precisely what Susskind conjectures for SYK: a low temperature description in terms of JT gravity, and a high temperature phase where the gravitational description is instead dS$_2$. While this is not conclusive evidence, it is suggestive.

Typically, the dual theory is located on a surface (or, in this case, a curve) in the weakly coupled region. In our construction the typical relationship is inverted, with the weakly coupled region in the deep interior of the spacetime. If a dual description proceeds along the usual lines, with the SYK-like theory living near (or at) the conformal boundary, then any concrete holographic proposal would have to account for the fact that gravity is strongly coupled in this region. On the other hand, the interior of spacetime in our model becomes a weakly coupled asymptotic region in a conformal frame where the dilaton kinetic term vanishes. So it may be that any relation between our model and SYK has a non-standard realization, localized along a point in the interior instead of near the conformal boundary. Since we have not yet explored the connection between SYK and our model, we chose to simply remark on the two observations above and not offer any further speculations.

The referee writes: How does your work compare to the "Centaur'' model in \href{https://arxiv.org/abs/1703.04622}{1703.04622} and its conclusions? There the authors also find a de Sitter realization embedded into anti-de Sitter whilst also having a strongly coupled region near the conformal boundary.

Our response: We would like to thank the referee for noticing this oversight. There was a reference in an earlier draft that was accidentally dropped during editing. We have replaced it in the text and added a comment that explains the differences between our model and the centaur geometry. The additions are in the first paragraph of section 6.

The referee writes: Above (118) you make an initial choice fixing $\tau_{\pm\pm}$, which corresponds to some heat bath. This, I believe, is the reason that you not find evaporation of any kind after applying shockwaves in the semi-classical case, see e.g. \href{https://arxiv.org/abs/2004.14944}{2004.14944}. Could you comment on this? Connected to the previous point, making a different choice for $\tau_{\pm\pm}$, it seems reasonable to get evaporation after performing a shockwave. Can you comment on this situation?

Our response: Indeed, since $T_{++} = 0$ there is no outgoing flux and therefore no evaporation. One could relax this by taking $T_{++}$ = (some positive flux) and then studying what happens. Our model is related (though not equivalent) to Jackiw-Teitelboim gravity, so one might find similar effects. This would be an interesting question to follow up on. However, our focus in this paper was determining whether an important feature of the classical model (the solution with horizon has lowest free energy) persists semiclassically. To that end, we focused on the backreaction to static solutions.

We listed the shockwave calcuation inspired by that paper as a viable generalization of our analysis in the third to last paragraph of section 6.

The referee writes: Is there any valid value $\eta$ that corresponds to a minimally coupled free scalar field in, e.g., 4d?

Our response: Yes, there is such a value, but assumptions elsewhere in the paper mean that this particular case is not accommodated by the model as currently formulated.

The value of $\eta$ corresponding to dimensionally reduced minimally coupled free scalars (from any higher dimension) is $\eta=1$. This is compatible with the inequality (103) and lies between the two values we considered, $\eta=0$ and $\eta=2$. However, the reduction of a 4D theory with minimally coupled scalar matter would give a linear coupling to the dilaton in (93), which we did not analyze in our work.

This and other dilaton-dependent matter couplings are an interesting generalization of our model, which we now mention in the third to last paragraph of section 6.

The referee writes: There have recently been some effort to use Quantum Extremal Surfaces or island formula to investigate unitarity of semi-classical de Sitter models using the Page curve. Could you study islands in your model, or do you expect that the strongly coupled region stops you from, e.g., attaching a bath region needed for collection of radiation?

Our response: We do not have a firm answer to this question. Based on our understanding, we expect that the presence of a strongly coupled region would not prevent attaching a bath needed for the collection of radiation. However, as stated above, our framework was focused on analyzing semiclassical effects for static backgrounds. So it may be that there are issues we are overlooking. Therefore we did not address this issue in the paper.

arXiv Update An updated version (v2) is now available on the arXiv. All changes from v1 to v2 are confined to section 6 and the references.

First paragraph: "...This is distinct from other constructions that yield a state-dependent cos- mological constant, at the price of introducing extra gauge fields [46–53]. It also differs from the “centaur geometry” studied in [54], which interpolates between AdS$_2$ in the UV and dS$_2$ in the IR. Unlike the centaur geometries, solutions in our model have constant curvature, whose value and sign are state-dependent."

Third to last paragraph: "In this paper we took an intrinsically 2D perspective. However, from the viewpoint of dimensional reduction a linear coupling of the matter field to the dilaton is required. Therefore, it could be rewarding to generalize our discussion to non-minimal coupling in (93). Another possible generalization is to relax the assumption of staticity in our treatment of backreaction. This would accommodate time dependent configurations such as the shockwaves in [55]."

Penultimate paragraph: "We conclude with an intriguing observation, and additional prospects for future re- search. ..."

References: [54] D. Anninos and D. Hofman, Infrared Realization of dS2 in AdS2, Class.Quant.Grav. 35, 085003 (2018), doi:10.1088/1361-6382/aab143, 1703.04622.

[55] T. J. Hollowood and S. P. Kumar, Islands and Page Curves for Evaporating Black Holes in JT Gravity, JHEP 08, 094 (2020), doi:10.1007/JHEP08(2020)094, 2004. 14944.

---

## Round 1 · Referee Report · Anonymous (Referee 2) · 2022-7-21

Report

This manuscript studies particular models of dilaton-gravity theories in two dimensions. These models are characterised by a particular dilaton potential that admits different solutions with constant curvature (positive, negative or zero). The curvature depends on the state considered, so with the same potential, the theory has solutions that are locally (A)dS_2 or Minkowski. This feature is new and certainly interesting. Another particular feature is that the potential depends on derivatives of the dilaton, so, in principle, it could be re-written in the more standard form by field redefinitions, but the authors claim that both models are still different (despite having the same EOMs) even at a classical level.

The manuscript is well organised and written and the physics is clearly relevant. They study the action and solutions first; then they compute the thermodynamics of the different solutions and analyse stability properties. In chapter 4 they add matter in the form of a massless scalar field and study shockwaves and in chapter 5 they do a semiclassical analysis including backreaction.

I believe the manuscript should be published in SciPost, but I do have some minor comments for the authors:

  • There are some comments throughout the text but it is not clear to me the motivation to study these models despite having this interesting property. Is is just a 2d curiosity? Or there are higher-dimensional reasons to study these models? Even in two dimensions, why should we study this particular family of potentials and not others?
  • This is probably me not properly understanding the text, but it seems to me that the solutions with dS_2 have a temperature that can be varied? I thought the temperature in dS is fixed and given by the dS radius. A similar question can be raised about the boundary conditions. In the manuscript the length of “a” circle is fixed to \beta, but where is this circle in the dS solutions and why it has to be interpreted as an inverse temperature? Similarly in the thermodynamic analysis there is an energy that is usually zero for dS for here it is free and given by M, for instance, in eq. (33).
  • The authors nicely change coordinates to conformal coordinates in chapter 4, that can be adapted easier to couple matter. The solutions to dilaton-gravity with a general potential coupled to a massless scalar field have been previously derived in 1811.08153, where the shockwave solutions are also presented for the case of AdS_2 and dS_2. It might be worth comparing these derivations. In a similar fashion to section 4.4, bubbles of dS_2 inside AdS_2 have been considered in, for instance, 2003.05460. I believe in the present manuscript the curvature inside the bubble does not change sign, but would it be possible to engineer such bubbles in the present theories?
  • Finally, I was going to ask about connections to holography, but the authors do mention it at the end of the discussion. They propose this models to be dual to the double-scaled SYK model at all temperatures, at low temperatures being AdS and at high temperature, dS. This seems a little bit weird to me because it would imply that SYK has a geometric description for all temperatures which I doubt it is the case. Moreover, in between AdS and dS, as the authors point out there is a planar case. Do we also have to think that the double-scaled SYK is dual to Minkowski in two dimensions, at some intermediate temperature? As it is a radical claim, I would like the authors to further comment on it.
  • validity: -
  • significance: -
  • originality: -
  • clarity: -
  • formatting: -
  • grammar: -

Author:  Robert McNees  on 2022-08-30  [id 2774]

(in reply to Report 2 on 2022-07-21)
Category:
answer to question

We thank the referee for their useful comments. A revised version of the paper has been submitted to the arXiv and should appear tomorrow. We address the points they raise below, in order.

  1. We study this particular family and no others because it has the unique property that all solutions have constant curvature, with state-dependent value and sign of curvature. It is this property that allows to view dS$_2$ as an excitation of AdS$_2$. We value this property so highly that it became the defining title of this work. Regarding higher dimensions, we take an intrinsically 2D point of view. It would be interesting if our 2d model appeared as some compactification of a higher-dimensional theory, but we leave this issue to future work.

  2. The periodicity $\beta$ of Euclidean time (the inverse of the temperature) is fixed uniquely by our boundary conditions, so it cannot be varied. This is discussed at the beginning of section 3, where the spectrum is summarized. The important point is that for a saddle point with horizon -- the only way dS$_2$ appears in the spectrum -- there is a unique solution that avoids a conical singularity. We refer to the value of the integration constant appearing in this smooth solution as "M." The value of M is fixed by the boundary condition $\beta$, and sets the value of the curvature.

  3. We thank the referee for pointing out reference 1811.08153, which we have added to our bibliography. In response to the question about engineering a bubble where the curvature changes sign, the curvature can only increase through this process. We have added some clarifying text after Eq. (91), explaining that the average null energy condition forbids a decrease in curvature. Thus, it is possible for a bubble of positive curvature to form inside a locally AdS$_2$ configuration, but it is not possible for a bubble of AdS$_2$ to nucleate inside a locally dS$_2$ solution.

  4. We do not claim in our paper that the model we study is holographically dual to double-scaled SYK. Rather, we note that pursuing holography would be of interest and make some suggestions for how the model would have to be modified (relaxing the condition that the boundary is a dilaton isosurface, for instance). One reason we find this an interesting avenue is that our model shares a feature in common with Susskind's holographic proposal: the transition to a dS$_2$ description at high temperatures.

List of Changes in updated version - Comment added after Eq. (91) explaining that bubble nucleation only increases the curvature, as a result of the averaged null energy condition. - Added reference [55] in the Discussion - Made a small edit in the conclusion, replacing "concrete" with "possible."

---

## Round 3 · Referee Report · Anonymous (Referee 1) · 2022-9-8

Report

I wrote report 1 and I am satisfied with the answers and changes of the authors.

---

## Round 3 · Referee Report · Anonymous (Referee 2) · 2022-9-27

Report

I appreciate the answers gave by the authors and the changes in the manuscript, so I am happy to suggest the manuscript for publication in SciPost.

---

## Round 3 · Author Response

This is a resubmission, after responding to comments and questions from referees.

---

## Round 3 · List of Changes

There have been two updates (v2 and v3) since the initial submission. The updates include changes in response to previous referee comments. The changes, by version, are as follows:

Version 2 - Section 6, First paragraph: "...This is distinct from other constructions that yield a state-dependent cos- mological constant, at the price of introducing extra gauge fields [46–53]. It also differs from the “centaur geometry” studied in [54], which interpolates between AdS in the UV and dS2 in the IR. Unlike the centaur geometries, solutions in our model have constant curvature, whose value and sign are state-dependent."

  • Section 6, Third to last paragraph: "In this paper we took an intrinsically 2D perspective. However, from the viewpoint of dimensional reduction a linear coupling of the matter field to the dilaton is required. Therefore, it could be rewarding to generalize our discussion to non-minimal coupling in (93). Another possible generalization is to relax the assumption of staticity in our treatment of backreaction. This would accommodate time dependent configurations such as the shockwaves in [55]."

  • Section 6, Penultimate paragraph: "We conclude with an intriguing observation, and additional prospects for future re- search. ..."

  • Added the following references: [54] D. Anninos and D. Hofman, Infrared Realization of dS2 in AdS2, Class.Quant.Grav. 35, 085003 (2018), doi:10.1088/1361-6382/aab143, 1703.04622.

[56] T. J. Hollowood and S. P. Kumar, Islands and Page Curves for Evaporating Black Holes in JT Gravity, JHEP 08, 094 (2020), doi:10.1007/JHEP08(2020)094, 2004. 14944.

Version 3 - Comment added after Eq. (91) explaining that bubble nucleation only increases the curvature, as a result of the averaged null energy condition. - Made a small clarifying edit in the conclusion, replacing "concrete" with "possible." - Added reference [55] in the Discussion [55] D. Anninos, D. A. Galante and D. M. Hofman, De Sitter horizons & holographic liquids, JHEP 07, 038 (2019), doi:10.1007/JHEP07(2019)038, 1811.08153.

---

## Editorial Decision

published